# MOMENTUM BENEFITS NON-IID FEDERATED LEARNING SIMPLY AND PROVABLY

**Ziheng Cheng**[*]
Peking University
alex-czh@stu.pku.edu.cn

**Xinmeng Huang**[*]
University of Pennsylvania
xinmengh@sas.upenn.edu

**Pengfei Wu**
Peking University
pengfeiwu1999@stu.pku.edu.cn

**Kun Yuan**[†]
Peking University
kunyuan@pku.edu.cn

## ABSTRACT

Federated learning is a powerful paradigm for large-scale machine learning, but it faces significant challenges due to unreliable network connections, slow communication, and substantial data heterogeneity across clients. FEDAVG and SCAFFOLD are two prominent algorithms to address these challenges. In particular, FEDAVG employs multiple local updates before communicating with a central server, while SCAFFOLD maintains a control variable on each client to compensate for "client drift" in its local updates. Various methods have been proposed to enhance the convergence of these two algorithms, but they either make impractical adjustments to the algorithmic structure or rely on the assumption of bounded data heterogeneity. This paper explores the utilization of momentum to enhance the performance of FEDAVG and SCAFFOLD. When all clients participate in the training process, we demonstrate that incorporating momentum allows FEDAVG to converge without relying on the assumption of bounded data heterogeneity even using a constant local learning rate. This is novel and fairly surprising as existing analyses for FEDAVG require bounded data heterogeneity even with diminishing local learning rates. In partial client participation, we show that momentum enables SCAFFOLD to converge provably faster without imposing any additional assumptions. Furthermore, we use momentum to develop new variance-reduced extensions of FEDAVG and SCAFFOLD, which exhibit state-of-the-art convergence rates. Our experimental results support all theoretical findings.

## 1 INTRODUCTION

Federated learning (FL) is a powerful paradigm for large-scale machine learning (Konečný et al., 2016; McMahan et al., 2017a). In situations where data and computational resources are dispersed among a diverse range of clients, including phones, tablets, sensors, hospitals, and other devices and agents, federated learning facilitates local data processing and collaboration among these clients (Kairouz et al., 2021). Consequently, a centralized model can be trained without transmitting decentralized data from clients directly to servers, thereby ensuring a fundamental level of privacy.

Federated learning encounters several significant challenges in algorithmic development. Firstly, the reliability and relatively slow nature of network connections between the server and clients pose obstacles to efficient communication during the training process. Secondly, the dynamic availability of only a small subset of clients for training at any given time demands strategies that can adapt to this variable environment. Lastly, the presence of substantial heterogeneity of non-iid data across different clients further complicates the training process.

FEDAVG (Konečný et al., 2016; McMahan et al., 2017a; Stich, 2019; Yu et al., 2019a; Lin et al., 2020; Wang & Joshi, 2021) emerges as a prevalent algorithm for FL, leveraging multiple stochastic gradient descent (SGD) steps within each client before communicating with a central server. While FEDAVG is readily implementable and succeeds in certain applications, its performance is notably

---

[*]Equal Contribution. This work is supported in part by National Natural Science Foundation of China (NSFC) Grant 12301392, 92370121, and 12288101

[†]Corresponding Author. Kun Yuan is also affiliated with National Engineering Labratory for Big Data Analytics and Applications, and AI for Science Institute, Beijing, China.

hindered by the presence of data heterogeneity, *i.e.*, non-iid clients, even when *all clients* participate in the training process (Li et al., 2019; Yang et al., 2021). To mitigate the influence of data heterogeneity, SCAFFOLD (Karimireddy et al., 2020b) maintains a control variable on each client to compensate for "client drift" in its local SGD updates, making convergence more robust to data heterogeneity and client sampling. Due to their practicality and effectiveness, FEDAVG and SCAFFOLD have become foundational algorithms in federated learning, leading to the development of numerous variants that cater to decentralized (Koloskova et al., 2020; Rizk et al., 2022; Nguyen et al., 2022; Alghunaim, 2023), compressed (Haddadpour et al., 2021; Reisizadeh et al., 2020; Mitra et al., 2021), asynchronous (Chen et al., 2020a;b; Xu et al., 2021a), and personalized (Fallah et al., 2020; Pillutla et al., 2022; Tan et al., 2022; T Dinh et al., 2020) federated learning scenarios.

Various methods have been proposed to enhance the convergence of FEDAVG, SCAFFOLD, and their variance-reduced[1] extensions. While exhibiting superior convergence rates, these approaches typically make impractical adjustments to algorithmic structures. For instance, STEM (Khanduri et al., 2021) requires increasing either the batch size or the number of local steps with algorithmic iterations. Similarly, CE-LSGD (Patel et al., 2022) and MIME (Karimireddy et al., 2020a) mandate computing a large-batch or even full-batch local gradient per round for each client. Additionally, FEDPROX (Li et al., 2020), FEDPD (Zhang et al., 2021), and FEDDYN (Durmus et al., 2021) rely on solving "local problems" to an extremely high precision. These adjustments may not align with the practical constraints in federated learning setups.

Furthermore, many of these algorithms, including FEDAVG, STEM, FEDPROX, MIME, and CE-SGD, still rely on the assumption of bounded data heterogeneity. When this assumption is violated, their theoretical analyses become invalid. While some algorithms, such as LED (Alghunaim, 2023) and VRL-SGD (Liang et al., 2019), can handle unbounded data heterogeneity, their convergence rates are not state-of-the-art, as stated in Table 1. These limitations motivate us to develop novel strategies that are easy to implement, robust to data heterogeneity, and exhibit superior convergence.

## 1.1 MAIN RESULTS AND CONTRIBUTIONS

This paper examines the utilization of *momentum* to enhance the performance of FEDAVG and SCAFFOLD. To ensure simplicity and practicality in implementations, we only introduce momentum to the local SGD steps, avoiding any inclusion of impractical elements, such as gradient computation of multiple minibatches or solving local problems to high precision. Remarkably, this straightforward approach effectively alleviates the necessity for stringent assumptions regarding bounded data heterogeneity, leading to noteworthy improvements in convergence rates. The main findings and contributions of this paper are summarized below.

First, when *all* clients participate in the training process:

- We show that incorporating momentum allows FEDAVG and its variance-reduced extension to *converge under unbounded data heterogeneity*, even using constant local learning rates. This is rather surprising as, to our knowledge, all existing analyses for FEDAVG, *e.g.*, Karimireddy et al. (2020b); Yang et al. (2021); Wang et al. (2020b) require bounded data heterogeneity even with diminishing local learning rates.

- We further establish that, by effectively removing the influence of data heterogeneity on convergence, momentum empowers FEDAVG and its variance-reduced extension with state-of-the-art convergence rates in the context of full client participation.

Second, when *partial* clients participate in the training process per iteration:

- The proposed SCAFFOLD-M that incorporates momentum into SCAFFOLD achieves provably faster convergence. To our knowledge, this is the *first* result that improves SCAFFOLD without imposing additional assumptions beyond those in Karimireddy et al. (2020b).

- We further introduce momentum to SCAFFOLD with variance reduction, obtaining the *first* variance-reduced FL algorithm that converges without bounded data heterogeneity. This method attains the state-of-the-art convergence rate in the context of partial client participation and unbounded data heterogeneity.

Tables 1 and 2 present a comprehensive comparison of the convergence rates and associated assumptions of prior algorithms and our proposed methods. It is observed that by simply adding momentum

---

[1]Throughout the paper, variance reduction refers to techniques aiming to mitigate the influence of within-client gradient stochasticity, as opposed to the inter-client data heterogeneity.

Table 1: The comparison of convergence rates of FL algorithms when **all clients** participate in training. Notation $L$ is the smoothness constant of objective functions, $\Delta = f(x^0) - \min_x f(x)$ is the initialization gap, $\sigma^2$ is the variance of gradient noises, $N$ is the number of clients, $K$ is the number of local steps per round, and $R$ is the number of communication rounds, $\zeta^2$ and $G$ are uniform bounds of data heterogeneity $(1/N)\sum_{1 \le i \le N} \|\nabla f_i(x) - \nabla f(x)\|^2$ and gradient norm $G := \sup_x \max_{1 \le i \le N} \|\nabla f_i(x)\|$ with $G^2 \gg \zeta^2$ typically. The "Assumptions" column lists all assumptions beyond Assumption 1 and 3.

| Algorithm | Convergence Rate $\mathbb{E}[\|\nabla f(\hat{x})\|^2] \lesssim$ | Assumptions |
|---|---|---|
| FEDAVG | | |
| (Yu et al., 2019b) | $\left(\frac{L\Delta\sigma^2}{NKR}\right)^{1/2} + \left(\frac{L\Delta G}{R}\right)^{2/3} + \frac{L\Delta}{R}$ | Bounded grad. |
| (Koloskova et al., 2020) | $\left(\frac{L\Delta\sigma^2}{NR}\right)^{1/2} + \left(\frac{L\Delta K\zeta}{R}\right)^{2/3} + \frac{L\Delta K}{R}$ | Bounded hetero. |
| (Karimireddy et al., 2020b) | $\left(\frac{L\Delta\sigma^2}{NKR}\right)^{1/2} + \left(\frac{L\Delta\zeta}{R}\right)^{2/3} + \frac{L\Delta}{R}$ | Bounded hetero. |
| (Yang et al., 2021) | $\left(\frac{L\Delta\sigma^2}{NKR}\right)^{1/2} + \frac{L\Delta}{R}$ | Bounded hetero.[1] |
| FEDCM[2] (Xu et al., 2021b) | $\left(\frac{L\Delta(\sigma^2+NKG^2)}{NKR}\right)^{1/2} + \left(\frac{L\Delta(\sigma/\sqrt{K}+G)}{R}\right)^{2/3}$ | Bounded grad. Bounded hetero. |
| LED (Alghunaim, 2023) | $\left(\frac{L\Delta\sigma^2}{NKR}\right)^{1/2} + \left(\frac{L\Delta\sigma}{\sqrt{K}R}\right)^{2/3} + \frac{L\Delta}{R}$ | – |
| VRL-SGD[2] (Liang et al., 2019) | $\left(\frac{L\Delta\sigma^2}{NKR}\right)^{1/2} + \left(\frac{L\Delta\sigma}{\sqrt{K}R}\right)^{2/3} + \frac{L\Delta}{R}$ | – |
| FEDAVG-M (Thm. 1) | $\left(\frac{L\Delta\sigma^2}{NKR}\right)^{1/2} + \frac{L\Delta}{R}$ | – |
| VARIANCE-REDUCTION | | |
| BVR-L-SGD (Murata & Suzuki, 2021) | $\left(\frac{L\Delta\sigma}{NKR}\right)^{2/3} + \frac{\sigma^2}{NKR} + \frac{L\Delta}{R}$ | Sample smooth $\mathcal{O}(K)$ minibatches[3] |
| CE-LSGD (Patel et al., 2022) | $\left(\frac{L\Delta\sigma}{NKR}\right)^{2/3} + \frac{\sigma^2}{NKR} + \frac{L\Delta}{R}$ | Sample smooth $\mathcal{O}(K)$ minibatches[3] |
| STEM (Khanduri et al., 2021) | $\frac{L\Delta+\sigma^2+\zeta^2}{(NKR)^{2/3}} + \frac{L\Delta}{R}$ | Sample smooth Bounded hetero. |
| FEDAVG-M-VR (Thm. 2) | $\left(\frac{L\Delta\sigma}{NKR}\right)^{2/3} + \frac{L\Delta}{R}$ | Sample smooth[4] |
| FEDAVG-M-VR (Thm. 15) | $\left(\frac{L\Delta\sigma}{NKR}\right)^{2/3} + \frac{\sigma^2}{NKR} + \frac{L\Delta}{R}$ | Sample smooth[4] |

[1] The local learning rate vanishes to zero when data heterogeneity is unbounded, *i.e.*, $\zeta \to \infty$.
[2] The works have not been published in peer-reviewed venues.
[3] A large number of minibatches are utilized on each client per communication round.
[4] The difference roots in the amount of batches used for initialization. See the values of $B$ in Thm. 15.

to local steps, FEDAVG, SCAFFOLD, and their variance-reduced variants all attain state-of-the-art convergence rates without resorting to further assumptions such as bounded data heterogeneity. We support our theoretical findings with extensive numerical experiments.

## 1.2 RELATED WORK

**FL with homogeneous clients.** FEDAVG is a well-known algorithm introduced by McMahan et al. (2017b) as a heuristic to enhance communication efficiency and data privacy in federated learning. Numerous subsequent studies have focused on analyzing its convergence under the assumption of homogeneous datasets, where clients are independent and identically distributed (iid) and all clients participate fully (Stich, 2019; Yu et al., 2019b; Wang & Joshi, 2021; Lin et al., 2020; Zhou & Cong, 2017). However, when dealing with heterogeneous clients and partial client participation, FEDAVG is found to be vulnerable to data heterogeneity because of the "client drift" effect (Karimireddy et al., 2020b; Yang et al., 2021; Wang et al., 2020b; Li et al., 2019).

**FL with heterogeneous clients.** Numerous research efforts are devoted to mitigating the impact of data heterogeneity in FL. For example, Li et al. (2020) propose FEDPROX, which introduces a proximal term to the objective function. Yang et al. (2021) utilize a two-sided learning rate approach, while Wang et al. (2020a) propose FEDNOVA, a normalized averaging method. Additionally, Zhang et al. (2021) presents FEDPD, which addresses data heterogeneity from a primal-dual optimization perspective. Notably, Karimireddy et al. (2020b) introduces SCAFFOLD, an effective algorithm that employs control variables to mitigate the influence of data heterogeneity and partial client participation. FEDGATE (Haddadpour et al., 2021) and LED (Alghunaim, 2023) are two recent effective algorithms that have alleviated the impact of data heterogeneity, utilizing gradient tracking (Xu et al.,

Table 2: The comparison of convergence rates of FL algorithms when **$S$ out of $N$ clients** participate in training per iteration. Notations are the same as those in Table 1.

| Algorithm | Convergence Rate $\mathbb{E}[\|\nabla f(\hat{x})\|^2] \lesssim$ | Assumptions |
|---|---|---|
| SCAFFOLD (Karimireddy et al., 2020b) | $\left(\frac{L\Delta\sigma^2}{SKR}\right)^{1/2} + \frac{L\Delta}{R}\left(\frac{N}{S}\right)^{2/3}$ | − |
| SCAFFOLD-M (Thm. 3) | $\left(\frac{L\Delta\sigma^2}{SKR}\right)^{1/2} + \frac{L\Delta}{R}\left(1 + \frac{N^{2/3}}{S}\right)$ | − |
| VARIANCE-REDUCTION | | |
| MIMELITEMVR[1] (Karimireddy et al., 2020a) | $\left(\frac{L\Delta(\sigma+\zeta)}{R}\right)^{2/3} + \frac{L\Delta+\sigma^2+\zeta^2}{R}$ | Sample smooth Noiseless grad. |
| MB-STORM (Patel et al., 2022) | $\left(\frac{L\Delta\sigma}{S\sqrt{K}R}\right)^{2/3} + \left(\frac{L\Delta\zeta}{SR}\right)^{2/3} + \frac{\zeta^2}{SR} + \frac{L\Delta}{R} + \frac{\sigma^2}{SKR}$ | Sample smooth Bounded hetero. $\mathcal{O}(K)$ minibatches[2] |
| CE-LSGD[1] (Patel et al., 2022) | $\left(\frac{L\Delta\sigma}{S\sqrt{K}R}\right)^{2/3} + \left(\frac{L\Delta\zeta}{SR}\right)^{2/3} + \frac{\zeta^2}{SR} + \frac{L\Delta}{R} + \frac{\sigma^2}{SKR}$ | Sample smooth Bounded hetero. $\mathcal{O}(K)$ minibatches[2] |
| SCAFFOLD-M-VR (Thm. 4) | $\left(\frac{L\Delta\sigma}{S\sqrt{K}R}\right)^{2/3} + \frac{L\Delta}{R}\left(1 + \frac{N^{1/2}}{S}\right)$ | Sample smooth[3] |
| SCAFFOLD-M-VR (Thm. 22) | $\left(\frac{L\Delta\sigma}{S\sqrt{K}R}\right)^{2/3} + \frac{L\Delta}{R}\left(1 + \frac{N^{1/2}}{S}\right) + \frac{\sigma^2}{SKR}$ | Sample smooth[3] |

[1] MIMELITEMVR and CE-LSGD consider the setting of streaming clients.
[2] A large number of minibatches are utilized on each client per communication round.
[3] The difference roots in the amount of batches used for initialization. See the values of $B$ in Thm. 22.

2015; Di Lorenzo & Scutari, 2016; Pu & Nedić, 2020; Xin et al., 2020; Alghunaim & Yuan, 2021; Huang et al., 2024) and exact-diffusion (Yuan et al., 2019; 2020; 2023) techniques, respectively.

**FL with momentum.** The momentum mechanism dates back to Nesterov's acceleration (Yurri, 2004) and Polyak's heavy-ball method (Polyak, 1964), which later flourishes in the stochastic optimization (Yan et al., 2018; Yu et al., 2019a; Liu et al., 2020) and other areas (Yuan et al., 2021; He et al., 2023b;a; Chen et al., 2023; Huang et al., 2024). Extensive research has explored incorporating momentum into FL (Reddi et al., 2021; Wang et al., 2020b; Karimireddy et al., 2020a; Khanduri et al., 2021; Patel et al., 2022; Das et al., 2022; Yu et al., 2019a; Xu et al., 2021b), and have demonstrated its impact on enhancing the empirical performance of FL methods (Wang et al., 2020b; Xu et al., 2021b; Reddi et al., 2021; Jin et al., 2022; Kim et al., 2022). However, whether momentum can offer *theoretical benefits* to FL, especially in mitigating the impact of data heterogeneity, remains unclear. This work demonstrates that momentum can benefit non-iid federated learning simply and provably. Notably, the utility of momentum is demonstrated in domains other than FL. For instance, Guo et al. (2021) proves that momentum can correct the bias experienced by the ADAM method, while recently Fatkhullin et al. (2023) shows that momentum can improve the error feedback technique in communication compression. The analysis presented in this work distinguishes from prior works including Guo et al. (2021); Fatkhullin et al. (2023) due to the unique challenges encountered in FL including multiple local updates, data heterogeneity, and partial client participation.

## 2 PROBLEM SETUP

This section formulates the problem of non-iid federated learning. Formally, we consider minimizing the following objective with the fewest number of client-server communication rounds:

$$\min_{x\in\mathbb{R}^d} \quad f(x) := \frac{1}{N}\sum_{i=1}^{N} f_i(x) \quad \text{where} \quad f_i(x) := \mathbb{E}_{\xi_i\sim\mathcal{D}_i}[F(x;\xi_i)].$$

Here, the random variable $\xi_i$ represents a local data point available at client $i$, while the function $f_i(x)$ denotes the non-convex local loss function associated with client $i$. This function takes expectation concerning the local data distribution $\mathcal{D}_i$. In practice, the local data distributions $\mathcal{D}_i$ among different clients typically differ from each other, resulting in the inequality $f_i(x) \neq f_j(x)$ for any pair of nodes $i$ and $j$. This phenomenon is commonly referred to as *data heterogeneity*. If all local clients were homogeneous, meaning that all local data samples follow the same distribution $\mathcal{D}$, we would have $f_i(x) = f_j(x)$ for any $i$ and $j$. In addition, throughout the paper, we assume that the function $f$ is bounded from below and possesses a global minimum $f^*$. To facilitate convergence analysis, we also introduce the following standard assumptions.

**Assumption 1** (STANDARD SMOOTHNESS)**.** *Each local objective $f_i$ is $L$-smooth, i.e., $\|\nabla f_i(x) - \nabla f_i(y)\| \leq L\|x - y\|$, for any $x, y \in \mathbb{R}^d$ and $1 \leq i \leq N$.*

**Assumption 2** (SAMPLE-WISE SMOOTHNESS)**.** *Each sample-wise objective $F(x;\xi)$ is $L$-smooth, i.e., $\|\nabla F(x;\xi_i) - \nabla F(y;\xi_i)\| \leq L\|x - y\|$ for any $x, y \in \mathbb{R}^d$, $1 \leq i \leq N$, and $\xi_i \overset{iid}{\sim} \mathcal{D}_i$.*

---

**Algorithm 1** FEDAVG-M: FEDAVG with momentum

---

**Require:** initial model $x^0$ and gradient estimate $g^0$, local learning rate $\eta$, global learning rate $\gamma$, momentum $\beta$
  **for** $r = 0, \cdots, R-1$ **do**
    **for** each client $i \in \{1, \ldots, N\}$ in parallel **do**
      Initialize local model $x_i^{r,0} = x^r$
      **for** $k = 0, \cdots, K-1$ **do**
        Compute $g_i^{r,k} = \beta \nabla F(x_i^{r,k}; \xi_i^{r,k}) + (1-\beta)g^r$             $\triangleright$ $\beta = 1$ implies FEDAVG
        Update local model $x_i^{r,k+1} = x_i^{r,k} - \eta g_i^{r,k}$
      **end for**
    **end for**
    Aggregate local updates $g^{r+1} = \frac{1}{\eta N K} \sum_{i=1}^{N} \left( x^r - x_i^{r,K} \right)$
    Update global model $x^{r+1} = x^r - \gamma g^{r+1}$
  **end for**

---

It is worth noting that Assumption 2 implies Assumption 1, which is typically used in variance-reduced algorithms, *e.g.*, Karimireddy et al. (2020a); Khanduri et al. (2021); Fang et al. (2018); Cutkosky & Orabona (2019). We will utilize either Assumption 1 or 2 in different algorithms. It is worth highlighting that, these are the *only* assumptions required for all our theoretical analyses.

**Assumption 3** (STOCHASTIC GRADIENT). *There exists $\sigma \geq 0$ such that for any $x \in \mathbb{R}^d$ and $1 \leq i \leq N$, $\mathbb{E}_{\xi_i}[\nabla F(x; \xi_i)] = \nabla f_i(x)$ and $\mathbb{E}_{\xi_i}[\|\nabla F(x; \xi_i) - \nabla f_i(x)\|^2] \leq \sigma^2$ where $\xi_i \overset{iid}{\sim} \mathcal{D}_i$.*

## 3 ACCELERATING FEDAVG WITH MOMENTUM

This section focuses on full client participation. We will introduce momentum to both FEDAVG and its variance-reduced extension. Furthermore, we will justify that the incorporation of momentum effectively mitigates the impact of data heterogeneity, leading to improved convergence rates.

### 3.1 FEDAVG WITH MOMENTUM

**Algorithm.** We introduce momentum to enhance the estimation of the stochastic gradient, resulting in the algorithm FEDAVG-M, as presented in Algorithm 1. In FEDAVG-M, the subscript $i$ represents the client index, while the superscripts $r$ and $k$ denote the outer loop index and inner local update index, respectively. The structure of FEDAVG-M remains identical to the vanilla FEDAVG, except for the inclusion of momentum in gradient computation (see highlight in Algorithm 1):

$$g_i^{r,k} = \beta \nabla F(x_i^{r,k}; \xi_i^{r,k}) + (1-\beta)g^r, \tag{1}$$

where $\beta \in [0,1]$ is the momentum coefficient, and $g^r$ represents a global gradient estimate updated in the outer loop $r$. It is important to note that FEDAVG-M will reduce to the vanilla FEDAVG when $\beta = 1$. Furthermore, FEDAVG-M is easy to implement, as it maintains the same algorithmic structure and incurs no additional uplink communication overhead compared to FEDAVG. Notably, no extra downlink commmunication cost is needed if clients store the last iterate model $x^r$ so that momentum $g^{r+1}$ can recovered through $(x^{r+1} - x^r)/\gamma$.

**Convergence property.** The inclusion of momentum in FEDAVG yields notable theoretical improvements. Firstly, it eliminates the need for the data heterogeneity assumption, also known as the gradient similarity assumption. The assumption can be expressed as

$$\frac{1}{N} \sum_{i=1}^{N} \|\nabla f_i(x) - \nabla f(x)\|^2 \leq \zeta^2, \quad \forall x \in \mathbb{R}^d \qquad \text{(Bounded data heterogeneity)}$$

where $\zeta^2$ measures the magnitude of data heterogeneity. By incorporating momentum, the above assumption is *no longer required* for the convergence analysis of FEDAVG. Secondly, momentum enables FEDAVG to converge at a state-of-the-art rate. These improvements are justified as follows:

**Theorem 1.** *Under Assumption 1 and 3, if we set $g^0 = 0$, $\beta$, $\gamma$, and $\eta$ as in (5), FEDAVG-M enjoys*

$$\frac{1}{R} \sum_{r=0}^{R-1} \mathbb{E}[\|\nabla f(x^r)\|^2] \lesssim \sqrt{\frac{L\Delta\sigma^2}{NKR}} + \frac{L\Delta}{R},$$

*where $\Delta \triangleq f(x^0) - \min_x f(x)$ and $\lesssim$ absorbs numeric numbers. See proof in Appendix B.1.*

**Comparison with** FEDAVG. Table 1 compares FEDAVG-M with prior algorithms when all clients participate in the training process. The results demonstrate that FEDAVG-M attains the most favorable convergence rate without relying on any assumption of data heterogeneity. Moreover, this rate matches the lower bound provided by Arjevani et al. (2019). Notably, a recent work (Huang et al., 2023) establishes the convergence of FEDAVG by relaxing the bounded data heterogeneity to a bound on $f^\star - \frac{1}{N}\sum_{i=1}^{N} f_i^\star$ where $f^\star \triangleq \min_x f(x)$ and $f_i^\star \triangleq f_i(x)$. However, their convergence does not benefit from local updates. Moreover, it still suffers from data heterogeneity $f^\star - \frac{1}{N}\sum_{i=1}^{N} f_i^\star$ and gets slow as the number of clients $N$ increases, resulting in a suboptimal rate.

**Comparison with** FEDCM. FEDAVG-M coincides with the FEDCM algorithm proposed by Xu et al. (2021b). However, our result outperforms that of Xu et al. (2021b) in several aspects. First, our convergence only utilizes the standard smoothness of objectives and gradient stochasticity while Xu et al. (2021b) additionally require bounded data heterogeneity and bounded gradients which are rarely valid in practice, suggesting the limitation of their result. Second, the convergence established by Xu et al. (2021b) is significantly weaker than ours and cannot even asymptotically approach the customary rate $O(1/\sqrt{NKR})$ in non-convex FL, as demonstrated by the results stated in Table 1.

**Constant local learning rate.** Based on Theorem 1, it can be inferred that when $R \gtrsim NKL\Delta/\sigma^2$, FEDAVG-M allows the utilization of *constant* local learning rate $\eta$ which does not necessarily decay as the number of communication rounds $R$ increases. This characteristic eases the tuning of the local learning rate and improves empirical performance. In contrast, many existing convergence results of FEDAVG necessitate the adoption of local learning rates that diminish as $R$ increases, as exemplified by *e.g.*, Yang et al. (2021); Li et al. (2019); Karimireddy et al. (2020b); Koloskova et al. (2020).

**Intuition on the effectiveness of momentum.** The momentum mechanism relies on an accumulated gradient estimate $g^r$, which is updated through $g^{r+1} = \frac{\beta}{NK}\sum_{i=1}^{N}\sum_{k=0}^{K-1} \nabla F(x_i^{r,k}; \xi_i^{r,k}) + (1-\beta)g^r$. While $g^r$ is a biased gradient estimate, it exhibits reduced variance due to its accumulation nature compared to a stochastic gradient $\nabla F(x_i^{r,k}; \xi_i^{r,k})$ computed with a single data minibatch. Importantly, by utilizing directions $\beta\nabla F(x_i^{r,k}; \xi_i^{r,k}) + (1-\beta)g^r$ for local updates, an "anchoring" effect is achieved, effectively mitigating the "client-drift" phenomenon. In the extreme case where $\beta = 0$, all clients remain synchronized in their local updates, eliminating the drift incurred by data heterogeneity in the vanilla FEDAVG. By appropriately tuning the coefficient $\beta$, FEDAVG-M maintains the same convergence rate as (Yang et al., 2021) while removing the requirement of data heterogeneity assumption utilized in their analysis.

## 3.2 VARIANCE-REDUCED FEDAVG WITH MOMENTUM

When each local loss function is further assumed to be sample-wise smooth (*i.e.*, Assumption 2), we can replace the local descent direction in Algorithm 1 with a variance-reduced momentum direction

$$g_i^{r,k} = \nabla F(x_i^{r,k}; \xi_i^{r,k}) + (1-\beta)(g^r - \nabla F(x^{r-1}; \xi_i^{r,k})) \tag{2}$$

to further enhance convergence, leading to variance-reduced FEDAVG with momentum, or FEDAVG-M-VR for short, see the detailed algorithm in Appendix B.2. The variable $x^{r-1}$ is the last-iterate global model maintained in the server. The construction of the variance-reduced direction (2) effectively mitigates the influence of within-client gradient noise and can be traced back to SARAH (Nguyen et al., 2017) and STORM (Cutkosky & Orabona, 2019) in stochastic optimization; more discussion can be found in Tan et al. (2022). Same as FEDAVG-M, turning off the variance-reduced momentum of FEDAVG-M-VR, *i.e.*, setting $\beta = 1$, recovers FEDAVG. FEDAVG-M-VR shares the same algorithmic structure and uplink communication workload as FEDAVG.

**Theorem 2.** *Under Assumption 2 and 3, if we take* $g^0 = \frac{1}{NB}\sum_{i=1}^{N}\sum_{b=1}^{B} \nabla F(x^0; \xi_i^b)^2$ *with* $\{\xi_i^b\}_{b=1}^{B} \overset{iid}{\sim} \mathcal{D}_i$ *and set* $\beta$, $\gamma$, $\eta$, *and* $B$ *as in* (8)*,* FEDAVG-M-VR *enjoys*

$$\frac{1}{R}\sum_{r=0}^{R-1} \mathbb{E}[\|\nabla f(x^r)\|^2] \lesssim \left(\frac{L\Delta\sigma}{NKR}\right)^{2/3} + \frac{L\Delta}{R}.$$

**Comparison with prior works.** FEDAVG-M-VR outperforms existing variance-reduced FL methods in convergence rate, as justified by the results listed in Table 1. Additionally, compared to BVR-L-SGD (Murata & Suzuki, 2021) and CE-LSGD (Patel et al., 2022), FEDAVG-M-VR conducts each local update using $1 + 1/K = O(1)$ minibatches on average, contrasting with the

---

[2]We use $B$ data minibatches per client to initialize the gradient estimate $g^0$ with small variance $\mathbb{E}[\|g^0 - \nabla f(x^0)\|^2]$, after which only one minibatch is utilized per local gradient computation. The same applies below.

---

**Algorithm 2** SCAFFOLD-M: SCAFFOLD with momentum

---

**Require:** initial model $x_0$, gradient estimator $g^0$, control variables $\{c_i^0\}_{i=1}^N$ and $c^0$, local learning rate $\eta$, global learning rate $\gamma$, momentum $\beta$

  **for** $r = 0, \cdots, R-1$ **do**

    Uniformly sample clients $\mathcal{S}_r \subseteq \{1, \cdots, N\}$ with $|\mathcal{S}_r| = S$

    **for** each client $i \in \mathcal{S}_r$ in parallel **do**

      Initialize local model $x_i^{r,0} = x^r$

      **for** $k = 0, \cdots, K-1$ **do**

        Compute $g_i^{r,k} = \beta(\nabla F(x_i^{r,k}; \xi_i^{r,k}) - c_i^r + c^r) + (1-\beta)g^r$   $\triangleright \beta = 1$ implies SCAFFOLD

        Update local model $x_i^{r,k+1} = x_i^{r,k} - \eta g_i^{r,k}$

      **end for**

      Update control variable $c_i^{r+1} := \frac{1}{K}\sum_{k=0}^{K-1} \nabla F(x_i^{r,k}; \xi_i^{r,k})$ (for $i \notin \mathcal{S}_r$, $c_i^{r+1} = c_i^r$)

    **end for**

    Aggregate local updates $g^{r+1} = \frac{1}{\eta S K}\sum_{i \in \mathcal{S}_r}\left(x^r - x_i^{r,K}\right)$

    Update global model $x^{r+1} = x^r - \gamma g^{r+1}$

    Update control variable $c^{r+1} = c^r + \frac{1}{N}\sum_{i \in \mathcal{S}_r}(c_i^{r+1} - c_i^r)$

  **end for**

---

$O(K)$ minibatches in BVR-L-SGD and CE-LSGD. Furthermore, in comparison to STEM (Khanduri et al., 2021), FEDAVG-M-VR does not rely on the assumption of bounded data heterogeneity.

Based on discussions in Sections 3.1 and 3.2, we demonstrate that FEDAVG-M and FEDAVG-M-VR, in the context of full client participation, can achieve the state-of-the-art convergence rate without resorting to any stronger assumption, *e.g.*, bounded data heterogeneity or impractical algorithmic structures such as a large number of minibatches in local gradient computation.

## 4 ACCELERATING SCAFFOLD WITH MOMENTUM

This section addresses the scenario where a random subset of clients participates in each training round. To tackle the challenges of partial participation, SCAFFOLD employs a control variable in each client to counteract the "client drift" effect during local updates. We will introduce momentum to both SCAFFOLD and its variance-reduced extension to gain better convergence results.

### 4.1 SCAFFOLD WITH MOMENTUM

**Algorithm.** We introduce momentum to enhance the estimation of the stochastic gradient, resulting in the newly proposed algorithm SCAFFOLD-M, outlined in Algorithm 2. In SCAFFOLD-M, $S$ clients are randomly selected from a pool of $N$ clients for each training iteration. The control variables $c_i$ and $c$ are maintained by the client and server, respectively. In SCAFFOLD, the local descent direction is given by $\nabla F(x_i^{r,k}; \xi_i^{r.k}) - c_i^r + c^r$. In contrast, SCAFFOLD-M incorporates momentum directions for local updates:

$$g_i^{r,k} = \beta(\nabla F(x_i^{r,k}; \xi_i^{r,k}) - c_i^r + c^r) + (1-\beta)g^r, \tag{3}$$

where $g^r$ represents the global stochastic gradient vector maintained by the server. It is worth noting that SCAFFOLD-M can reduce to SCAFFOLD by setting $\beta = 1$.

**Convergence property.** Our momentum yields notable theoretical improvements to SCAFFOLD:

**Theorem 3.** *Under Assumption 1 and 3, if we take $g^0 = 0$, $c_i^0 = \frac{1}{B}\sum_{b=1}^B \nabla F(x^0; \xi_i^b)$ with $\{\xi_i^b\}_{b=1}^B \overset{iid}{\sim} \mathcal{D}_i$, $c^0 = \frac{1}{N}\sum_{i=1}^N c_i^0$ and set $\beta$, $\gamma$, $\eta$, and $B$ as in (9), SCAFFOLD-M enjoys*

$$\frac{1}{R}\sum_{r=0}^{R-1} \mathbb{E}[\|\nabla f(x^r)\|^2] \lesssim \sqrt{\frac{L\Delta\sigma^2}{SKR}} + \frac{L\Delta}{R}\left(1 + \frac{N^{2/3}}{S}\right).$$

**Comparison with SCAFFOLD.** Compared to SCAFFOLD, SCAFFOLD-M exhibits provably faster convergence under partial participation, as justified in the comparison in Table 2. Specifically, when the gradients are noiseless (*i.e.*, $\sigma^2 = 0$), achieving the same level of stationarity $\mathbb{E}[\|\nabla f(\hat{x})\|^2]$ requires a ratio, between SCAFFOLD-M and SCAFFOLD, of communication rounds:

$$\frac{1 + N^{2/3}/S}{(N/S)^{2/3}} = \left(\frac{S}{N}\right)^{2/3} + \frac{1}{S^{1/3}}.$$

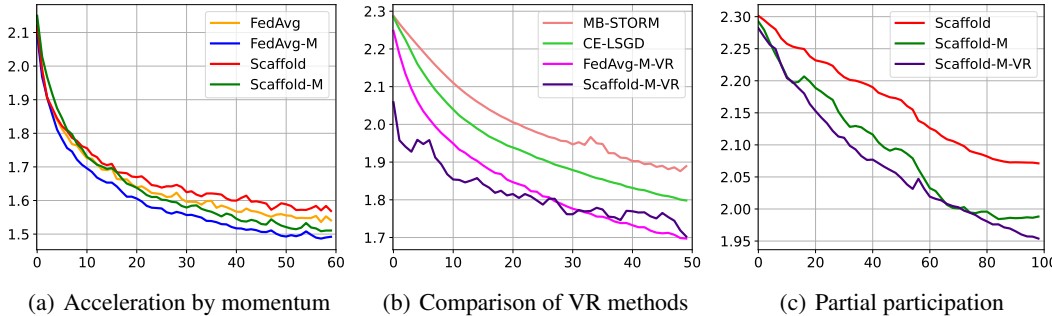

(a) Acceleration by momentum  (b) Comparison of VR methods  (c) Partial participation

Figure 1: Test loss of three-layer MLP versus the number of communication rounds

Thus, if $S \asymp N^{2/3}$, SCAFFOLD-M achieves up to $N^{2/9}$ times improvement in comparison to the vanilla SCAFFOLD, when aiming for the same level of stationarity. This improvement is significant as $N$, the number of clients in FL, is typically large. It is also worth highlighting that prior to our SCAFFOLD-M, SCAFFOLD was the only known non-iid FL method, to the best of our knowledge, that is robust to both unbounded data heterogeneity and partial client sampling, and capable of attaining linear speedup without relying on impractical algorithmic structures. The development of SCAFFOLD-M provides an alternative and superior choice.

## 4.2 VARIANCE-REDUCED SCAFFOLD WITH MOMENTUM

Similar to FEDAVG-M-VR, when the loss functions further enjoy the sample-wise smoothness property, we can obtain SCAFFOLD-M-VR by replacing momentum directions in Algorithm 2 with variance-reduced momentum directions

$$g_i^{r,k} = \nabla F(x_i^{r,k}; \xi_i^{r,k}) - \beta(c_i^r - c^r) + (1-\beta)(g^r - \nabla F(x^{r-1}; \xi_i^{r,k})).$$

The detailed algorithm is in Appendix C.2, and the convergence is shown below.

**Theorem 4.** *Under Assumption 2 and 3, if we take $c_i^0 = \frac{1}{B}\sum_{b=1}^B \nabla F(x^0; \xi_i^b)$ with $\{\xi_i^b\}_{b=1}^B \overset{iid}{\sim} \mathcal{D}_i$, $g^0 = c^0 = \frac{1}{N}\sum_{i=1}^N c_i^0$ and set $\beta$, $\gamma$, $\eta$, and $B$ as in* (11)*, SCAFFOLD-M-VR enjoys*

$$\frac{1}{R}\sum_{r=0}^{R-1} \mathbb{E}[\|\nabla f(x^r)\|^2] \lesssim \left(\frac{L\Delta\sigma}{S\sqrt{K}R}\right)^{2/3} + \frac{L\Delta}{R}\left(1 + \frac{N^{1/2}}{S}\right).$$

**Comparison with variance-reduced methods.** SCAFFOLD-M-VR outperforms all existing variance-reduced federated learning methods under partial participation in terms of convergence rate when data heterogeneity is severe (*i.e.*, $\zeta^2$ is large), see results listed in Table 2. Moreover, SCAFFOLD-M-VR has the following additional advantages. Compared to MIMELITEMVR (Karimireddy et al., 2020a), SCAFFOLD-M-VR does not need access to noiseless (full-batch) local gradients per iteration. Compared to MB-STORM (Patel et al., 2022) and CE-LSGD (Patel et al., 2022), SCAFFOLD-M-VR does not require bounded data heterogeneity and conducts each local update using $1 + 1/K = O(1)$ minibatches on average, instead of $\mathcal{O}(K)$.

Based on Sections 4.1 and 4.2, we demonstrate that SCAFFOLD-M and SCAFFOLD-M-VR, in the context of partial client participation, can achieve state-of-the-art convergence rates without resorting to any stronger assumption, *e.g.*, bounded data heterogeneity or impractical algorithmic structures such as a large number of minibatches in local gradient computation.

## 5 EXPERIMENTS

We present experiments on the CIFAR-10 dataset (Krizhevsky & Hinton, 2009) with two neural networks (three-layer MLP, ResNet-18) to justify the efficacy of our proposed algorithms. We evaluate them along with baselines including FEDAVG (Konečný et al., 2016), SCAFFOLD (Karimireddy et al., 2020b), MB-STORM, CE-LSGD (Patel et al., 2022). Parameters (such as learning rates) in our implementation are set by grid search. We defer more experimental details and results (*e.g.*, investigating the impact of momentum value $\beta$, setups with large $N$) to Appendix D.

### 5.1 MLP EXPERIMENTS

The MLP experiments involve $K = 32$ local updates and $N = 10$ clients with data generated via the Dirichlet distribution (Hsu et al., 2019) with a parameter of $0.5$ and $0.2$ for full and partial client participation, respectively (small parameter value implies severe heterogeneity).

Firstly, we compare the performance of FEDAVG-M and SCAFFOLD-M with their momentum-free counterparts, namely the vanilla FEDAVG and SCAFFOLD, under full client participation.

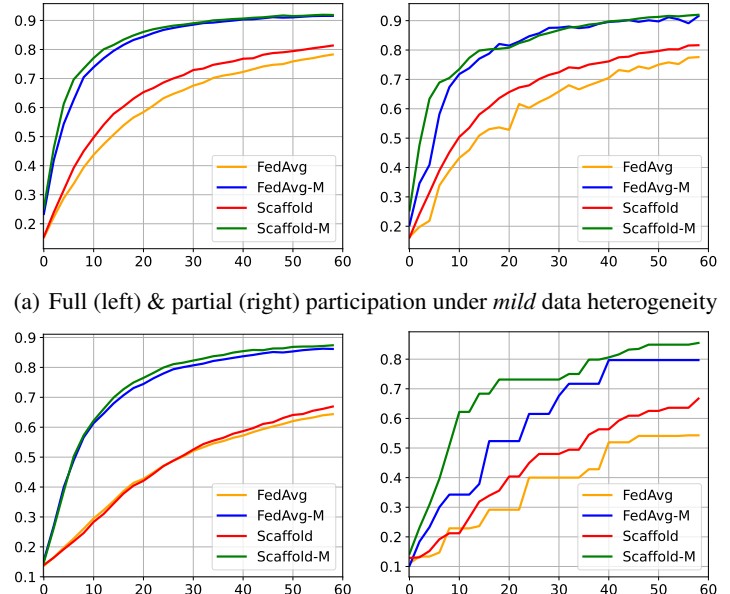

(a) Full (left) & partial (right) participation under *mild* data heterogeneity

(b) Full (left) & partial (right, best-so-far) participation under *severe* heterogeneity

Figure 2: Test accuracy of ResNet18 versus the number of communication rounds

The results are presented in Figure 1(a), in which it can be observed that incorporating momentum significantly accelerates the convergence of both FEDAVG and SCAFFOLD.

Secondly, we compare four momentum-based variance-reduced methods: MINIBATCH-STORM, CE-LSGD, FEDAVG-M-VR (our Algorithm 3), and SCAFFOLD-M-VR (our Algorithm 4), under full client participation. The comparison is illustrated in Figure 1(b). Our proposed methods outperform MINIBATCH-STORM and CE-LSGD with substantial margins.

Lastly, we investigate the case of partial client participation with $S = 1$ and compare the performance of SCAFFOLD-M and SCAFFOLD-M-VR with vanilla SCAFFOLD. The results are presented in Figure 1(c). Once again, we observe that the introduction of momentum leads to significant improvements even when only a few clients participate in each round of training.

## 5.2 RESNET18 EXPERIMENTS

We further compare the above algorithms with a larger model: ResNet18 (He et al., 2016) under varying data heterogeneity by setting the parameter of Dirichlet distribution as 0.5 and 0.1, respectively, where a small parameter value suggests severe data heterogeneity. The experiment involves $N = 10$ clients and $K = 16$ local updates. We set $S = 2$ in partial client participation.

Figure 2(a) reports the test accuracy of full and partial client participation under mild data heterogeneity while Figure 2(b) presents the counterparts under severe data heterogeneity, where the bottom right one is smoothed by plotting the best-so-far result. Again, we observe that FEDAVG-M and SCAFFOLD-M significantly outperform the vanilla FEDAVG and SCAFFOLD. Moreover, for ResNet18 and severe data heterogeneity, FEDAVG-M and SCAFFOLD-M exhibit notably greater advantages over their momentumless counterparts than for MLP scenarios under milder data heterogeneity. The observation demonstrates amplified advantages of the introduced momentum in larger models and severely heterogeneous data, which is aligned with our theoretical predictions and suggests the promising utility of our proposed methods in real-world applications.

## 6 CONCLUSION

We propose momentum variants of FEDAVG and SCAFFOLD under various client participation situations and objectives' smoothness. All the momentum variants make simple and practical adjustments to FEDAVG and SCAFFOLD yet obtain state-of-the-art performance among their peers, especially under severe data heterogeneity or small gradient variance. In particular, FEDAVG-M converges under unbounded data heterogeneity and admits constant local learning rates, giving the *first* neat convergence for FEDAVG-type methods; SCAFFOLD-M is the *first* FL method that outperforms SCAFFOLD unconditionally. Experiments conducted support our theoretical findings.

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

## A    PRELIMINARIES OF PROOFS

Let $\mathcal{F}^0 = \emptyset$ and $\mathcal{F}_i^{r,k} := \sigma(\{x_i^{r,j}\}_{0 \leq j \leq k} \cup \mathcal{F}^r)$ and $\mathcal{F}^{r+1} := \sigma(\cup_i \mathcal{F}_i^{r,K})$ for all $r \geq 0$ where $\sigma(\cdot)$ indicates the $\sigma$-algebra. Let $\mathbb{E}_r[\cdot] := \mathbb{E}[\cdot | \mathcal{F}^r]$ be the expectation, conditioned on the filtration $\mathcal{F}^r$, with respect to the random variables $\{\mathcal{S}^r, \{\xi_i^{r,k}\}_{1 \leq i \leq N, 0 \leq k < K}\}$ in the $r$-th iteration. We also use $\mathbb{E}[\cdot]$ to denote the global expectation over all randomness in algorithms. Through out the proofs, we use $\sum_i$ to represent the sum over $i \in \{1, \ldots, N\}$, while $\sum_{i \in \mathcal{S}^r}$ denotes the sum over $i \in \mathcal{S}^r$. Similarly, we use $\sum_k$ to represent the sum of $k \in \{0, \ldots, K-1\}$. For all $r \geq 0$, we define the following auxiliary variables to facilitate proofs:

$$\mathcal{E}_r := \mathbb{E}[\|\nabla f(x^r) - g^{r+1}\|^2],$$

$$U_r := \frac{1}{NK} \sum_i \sum_k \mathbb{E}[\|x_i^{r,k} - x^r\|]^2,$$

$$\zeta_i^{r,k} := \mathbb{E}[x_i^{r,k+1} - x_i^{r,k} | \mathcal{F}_i^{r,k}],$$

$$\Xi_r := \frac{1}{N} \sum_{i=1}^N \mathbb{E}[\|\zeta_i^{r,0}\|^2],$$

$$V_r := \frac{1}{N} \sum_{i=1}^N \mathbb{E}[\|c_i^r - \nabla f_i(x^{r-1})\|^2].$$

We remark that quantity $V_r$ is only used in the analysis of SCAFFOLD-based algorithms. Throughout the appendix, we let $\Delta := f(x^0) - f^*$, $G_0 := \frac{1}{N} \sum_i \|\nabla f_i(x^0)\|^2$, $x^{-1} := x^0$ and $\mathcal{E}_{-1} := \mathbb{E}[\|\nabla f(x^0) - g^0\|^2]$. We will use the following foundational lemma for all our algorithms.

**Lemma 5.** *Under Assumption 1, if $\gamma L \leq \frac{1}{24}$, the following holds all $r \geq 0$:*

$$\mathbb{E}[f(x^{r+1})] \leq \mathbb{E}[f(x^r)] - \frac{11\gamma}{24}\mathbb{E}[\|\nabla f(x^r)\|^2] + \frac{13\gamma}{24}\mathcal{E}_r.$$

*Proof.* Since $f$ is $L$-smooth, we have

$$f(x^{r+1}) \leq f(x^r) + \langle \nabla f(x^r), x^{r+1} - x^r \rangle + \frac{L}{2}\|x^{r+1} - x^r\|^2$$

$$= f(x^r) - \gamma\|\nabla f(x^r)\|^2 + \gamma\langle \nabla f(x^r), \nabla f(x^r) - g^{r+1}\rangle + \frac{L\gamma^2}{2}\|g^{r+1}\|^2.$$

Since $x^{r+1} = x^r - \gamma g^{r+1}$, using Young's inequality, we further have

$$f(x^{r+1})$$

$$\leq f(x^r) - \frac{\gamma}{2}\|\nabla f(x^r)\|^2 + \frac{\gamma}{2}\|\nabla f(x^r) - g^{r+1}\|^2 + L\gamma^2(\|\nabla f(x^r)\|^2 + \|\nabla f(x^r) - g^{r+1}\|^2)$$

$$\leq f(x^r) - \frac{11\gamma}{24}\|\nabla f(x^r)\|^2 + \frac{13\gamma}{24}\|\nabla f(x^r) - g^{r+1}\|^2,$$

where the last inequality is due to $\gamma L \leq \frac{1}{24}$. Taking the global expectation completes the proof. $\quad\square$

To handle local updates and client sampling, we will also use the following technical lemmas.

**Lemma 6** (Karimireddy et al. (2020b)). *Suppose $\{X_1, \cdots, X_\tau\} \subset \mathbb{R}^d$ be random variables that are potentially dependent. If their marginal means and variances satisfy $\mathbb{E}[X_i] = \mu_i$ and $\mathbb{E}[\|X_i - \mu_i\|^2] \leq \sigma^2$, then it holds that*

$$\mathbb{E}\left[\left\|\sum_{i=1}^\tau X_i\right\|^2\right] \leq \left\|\sum_{i=1}^\tau \mu_i\right\|^2 + \tau^2\sigma^2.$$

*If they are correlated in the Markov way such that $\mathbb{E}[X_i | X_{i-1}, \cdots X_1] = \mu_i$ and $\mathbb{E}[\|X_i - \mu_i\|^2 \mid \mu_i] \leq \sigma^2$, i.e., the variables $\{X_i - \mu_i\}$ form a martingale. Then the following tighter bound holds:*

$$\mathbb{E}\left[\left\|\sum_{i=1}^\tau X_i\right\|^2\right] \leq 2\mathbb{E}\left[\left\|\sum_{i=1}^\tau \mu_i\right\|^2\right] + 2\tau\sigma^2.$$

**Lemma 7.** *Given vectors $v_1, \cdots, v_N \in \mathbb{R}^d$ and $\bar{v} = \frac{1}{N}\sum_{i=1}^{N} v_i$, if we sample $\mathcal{S} \subset \{1, \cdots, N\}$ uniformly randomly such that $|\mathcal{S}| = S$, then it holds that*

$$\mathbb{E}\left[\left\|\frac{1}{S}\sum_{i\in\mathcal{S}} v_i\right\|^2\right] = \|\bar{v}\|^2 + \frac{N-S}{S(N-1)}\frac{1}{N}\sum_{i=1}^{N}\|v_i - \bar{v}\|^2.$$

*Proof.* Letting $\mathbb{1}\{i \in \mathcal{S}\}$ be the indicator for the event $i \in \mathcal{S}_r$, we prove this lemma by direct calculation as follows:

$$\mathbb{E}\left[\left\|\frac{1}{S}\sum_{i\in\mathcal{S}} v_i\right\|^2\right] = \mathbb{E}\left[\left\|\frac{1}{S}\sum_{i=1}^{N} v_i \mathbb{1}\{i \in \mathcal{S}\}\right\|^2\right]$$

$$= \frac{1}{S^2}\mathbb{E}\left[\left(\sum_i \|v_i\|^2 \mathbb{1}\{i \in \mathcal{S}\} + 2\sum_{i<j} v_i^\top v_j \mathbb{1}\{i,j \in \mathcal{S}\}\right)\right]$$

$$= \frac{1}{SN}\sum_{i=1}^{N}\|v_i\|^2 + \frac{1}{S^2}\frac{S(S-1)}{N(N-1)}2\sum_{i<j} v_i^\top v_j$$

$$= \frac{1}{SN}\sum_{i=1}^{N}\|v_i\|^2 + \frac{1}{S^2}\frac{S(S-1)}{N(N-1)}\left(\left\|\sum_{i=1}^{N} v_i\right\|^2 - \sum_{i=1}^{N}\|v_i\|^2\right)$$

$$= \frac{N-S}{S(N-1)}\frac{1}{N}\sum_{i=1}^{N}\|v_i\|^2 + \frac{N(S-1)}{S(N-1)}\|\bar{v}\|^2$$

$$= \frac{N-S}{S(N-1)}\frac{1}{N}\sum_{i=1}^{N}\|v_i - \bar{v}\|^2 + \|\bar{v}\|^2.$$

$\square$

In the following subsections, we present complete proofs of our main results. For FEDAVG-M and SCAFFOLD-M, our proofs only rely on Assumption 1 and 3, while for FEDAVG-M-VR and SCAFFOLD-M-VR, our proofs rely on Assumption 2 and 3.

## B    FEDAVG WITH MOMENTM

### B.1    FEDAVG-M

In this subsection, we present the proofs for the FEDAVG-M algorithm.

**Lemma 8.** *If* $\gamma L \leq \frac{\beta}{6}$*, the following holds for* $r \geq 1$*:*

$$\mathcal{E}_r \leq \left(1 - \frac{8\beta}{9}\right)\mathcal{E}_{r-1} + \frac{4\gamma^2 L^2}{\beta}\mathbb{E}[\|\nabla f(x^{r-1})\|^2] + \frac{2\beta^2\sigma^2}{NK} + 4\beta L^2 U_r.$$

*Additionally, it holds for* $r = 0$ *that*

$$\mathcal{E}_0 \leq (1 - \beta)\mathcal{E}_{-1} + \frac{2\beta^2\sigma^2}{NK} + 4\beta L^2 U_0.$$

*Proof.* For $r \geq 1$,

$$\mathcal{E}_r = \mathbb{E}[\|\nabla f(x^r) - g^{r+1}\|^2]$$

$$= \mathbb{E}\left[\left\|(1 - \beta)(\nabla f(x^r) - g^r) + \beta\left(\nabla f(x^r) - \frac{1}{NK}\sum_i\sum_k\nabla F(x_i^{r,k};\xi_i^{r,k})\right)\right\|^2\right]$$

$$= \mathbb{E}\left[\|(1 - \beta)(\nabla f(x^r) - g^r)\|^2\right] + \beta^2\mathbb{E}\left[\left\|\nabla f(x^r) - \frac{1}{NK}\sum_{i,k}\nabla F(x_i^{r,k};\xi_i^{r,k})\right\|^2\right]$$

$$+ 2\beta\mathbb{E}\left[\left\langle(1 - \beta)(\nabla f(x^r) - g^r), \nabla f(x^r) - \frac{1}{NK}\sum_{i,k}\nabla f(x_i^{r,k})\right\rangle\right].$$

Note that $\{\nabla F(x_i^{r,k};\xi_i^{r,k})\}_{0 \leq k < K}$ are sequentially correlated. Applying the AM-GM inequality and Lemma 6, we have

$$\mathcal{E}_r \leq \left(1 + \frac{\beta}{2}\right)\mathbb{E}[\|(1 - \beta)(\nabla f(x^r) - g^r)\|^2] + 2\beta L^2 U_r + 2\beta^2\left(\frac{\sigma^2}{NK} + L^2 U_r\right).$$

Using the AM-GM inequality again and Assumption 1, we have

$$\mathcal{E}_r \leq (1 - \beta)^2\left(1 + \frac{\beta}{2}\right)\left[\left(1 + \frac{\beta}{2}\right)\mathcal{E}_{r-1} + \left(1 + \frac{2}{\beta}\right)L^2\mathbb{E}[\|x^r - x^{r-1}\|^2]\right] + \frac{2\beta^2\sigma^2}{NK} + 4\beta L^2 U_r$$

$$\leq (1 - \beta)\mathcal{E}_{r-1} + \frac{2}{\beta}L^2\mathbb{E}[\|x^r - x^{r-1}\|^2] + \frac{2\beta^2\sigma^2}{NK} + 4\beta L^2 U_r$$

$$\leq \left(1 - \frac{8\beta}{9}\right)\mathcal{E}_{r-1} + 4\frac{\gamma^2 L^2}{\beta}\mathbb{E}[\|\nabla f(x^{r-1})\|^2] + \frac{2\beta^2\sigma^2}{NK} + 4\beta L^2 U_r,$$

where we plug in $\|x^r - x^{r-1}\|^2 \leq 2\gamma^2(\|\nabla f(x^{r-1})\|^2 + \|g^r - \nabla f(x^{r-1})\|^2)$ and use $\gamma L \leq \frac{\beta}{6}$ in the last inequality. Similarly for $r = 0$,

$$\mathcal{E}_0 \leq \left(1 + \frac{\beta}{2}\right)\mathbb{E}[\|(1 - \beta)(\nabla f(x^0) - g^0)\|^2] + 2\beta L^2 U_0 + 2\beta^2\left(\frac{\sigma^2}{NK} + L^2 U_0\right)$$

$$\leq (1 - \beta)\mathcal{E}_{-1} + \frac{2\beta^2\sigma^2}{NK} + 4\beta L^2 U_0.$$

$\square$

**Lemma 9.** *If* $\eta LK \leq \frac{1}{\beta}$*, the following holds for* $r \geq 0$*:*

$$U_r \leq 2eK^2\Xi_r + K\eta^2\beta^2\sigma^2(1 + 2K^3 L^2\eta^2\beta^2).$$

*Proof.* Recall that $\zeta_i^{r,k} := \mathbb{E}[x_i^{r,k+1} - x_i^{r,k}|\mathcal{F}_i^{r,k}] = -\eta\left((1 - \beta)g^r + \beta\nabla f_i(x_i^{r,k})\right)$. Then we have

$$\mathbb{E}[\|\zeta_i^{r,j} - \zeta_i^{r,j-1}\|^2] \leq \eta^2 L^2\beta^2\mathbb{E}[\|x_i^{r,j} - x_i^{r,j-1}\|^2]$$

$$\leq \eta^2 L^2\beta^2(\eta^2\beta^2\sigma^2 + \mathbb{E}[\|\zeta_i^{r,j-1}\|^2]).$$

For any $1 \leq j \leq k - 1 \leq K - 2$, using $\eta L \leq \frac{1}{\beta K} \leq \frac{1}{\beta(k+1)}$, we have

$$\mathbb{E}[\|\zeta_i^{r,j}\|^2] \leq \left(1 + \frac{1}{k}\right) \mathbb{E}[\|\zeta_i^{r,j-1}\|^2] + (1 + k)\mathbb{E}[\|\zeta_i^{r,j} - \zeta_i^{r,j-1}\|^2]$$

$$\leq \left(1 + \frac{2}{k}\right) \mathbb{E}[\|\zeta_i^{r,j-1}\|^2] + (k + 1)L^2\eta^4\beta^4\sigma^2$$

$$\leq e^2\mathbb{E}[\|\zeta_i^{r,0}\|^2] + 4k^2L^2\eta^4\beta^4\sigma^2,$$

where the last inequality is by unrolling the recursive bound and using $\left(1 + \frac{2}{k}\right)^k \leq e^2$. By Lemma 6, it holds that for $k \geq 2$,

$$\mathbb{E}[\|x_i^{r,k} - x^r\|^2] \leq 2\mathbb{E}\left[\left\|\sum_{j=0}^{k-1} \zeta_i^{r,j}\right\|^2\right] + 2k\eta^2\beta^2\sigma^2$$

$$\leq 2k \sum_{j=0}^{k-1} \mathbb{E}[\|\zeta_i^{r,k}\|^2] + 2k\eta^2\beta^2\sigma^2$$

$$\leq 2e^2k^2\mathbb{E}[\|\zeta_i^{r,0}\|^2] + 2k\eta^2\beta^2\sigma^2(1 + 4k^3L^2\eta^2\beta^2).$$

This is also valid for $k = 0, 1$. Summing up over $i$ and $k$ finishes the proof. $\square$

**Lemma 10.** *If $288e(\eta KL)^2((1 - \beta)^2 + e(\beta\gamma LR)^2) \leq 1$, then it holds for $r \geq 0$ that*

$$\sum_{r=0}^{R-1} \Xi_r \leq \frac{1}{72eK^2L^2} \sum_{r=-1}^{R-2} (\mathcal{E}_r + \mathbb{E}[\|\nabla f(x^r)\|^2]) + 2\eta^2\beta^2eRG_0.$$

*Proof.* Note that $\zeta_i^{r,0} = -\eta((1 - \beta)g^r + \beta\nabla f_i(x^r))$,

$$\frac{1}{N} \sum_{i=1}^{N} \|\zeta_i^{r,0}\|^2 \leq 2\eta^2 \left((1 - \beta)^2\|g^r\|^2 + \beta^2 \frac{1}{N} \sum_{i=1}^{N} \|\nabla f_i(x^r)\|^2\right).$$

Using Young's inequality, we have for any $q > 0$ that

$$\mathbb{E}[\|\nabla f_i(x^r)\|^2] \leq (1 + q)\mathbb{E}[\|\nabla f_i(x^{r-1})\|^2] + (1 + q^{-1})L^2\mathbb{E}[\|x^r - x^{r-1}\|^2]$$

$$\leq (1 + q)\mathbb{E}[\|\nabla f_i(x^{r-1})\|^2] + 2(1 + q^{-1})\gamma^2L^2(\mathcal{E}_{r-1} + \mathbb{E}[\|\nabla f(x^{r-1})\|^2])$$

$$\leq (1 + q)^r\mathbb{E}[\|\nabla f_i(x^0)\|^2] + \frac{2}{q}\gamma^2L^2 \sum_{j=0}^{r-1} (\mathcal{E}_j + \mathbb{E}[\|\nabla f(x^j)\|^2])(1 + q)^{r-j}.$$

Take $q = \frac{1}{r}$ and we have

$$\mathbb{E}[\|\nabla f_i(x^r)\|^2] \leq e\mathbb{E}[\|\nabla f_i(x^0)\|^2] + 2e(r + 1)\gamma^2L^2 \sum_{j=0}^{r-1} (\mathcal{E}_j + \mathbb{E}[\|\nabla f(x^j)\|^2]). \tag{4}$$

Note that this inequality is valid for $r = 0$. Therefore, using (4), we have

$$
\sum_{r=0}^{R-1} \Xi_r \leq \sum_{r=0}^{R-1} 2\eta^2 \mathbb{E}\left[(1-\beta)^2\|g^r\|^2 + \beta^2 \frac{1}{N}\sum_{i=1}^{N}\|\nabla f_i(x^r)\|^2\right]
$$

$$
\leq \sum_{r=0}^{R-1} 2\eta^2 \left(2(1-\beta)^2(\mathcal{E}_{r-1} + \mathbb{E}[\|\nabla f(x^{r-1})\|^2]) + \beta^2 \frac{1}{N}\sum_{i=1}^{N}\mathbb{E}[\|\nabla f_i(x^r)\|^2]\right)
$$

$$
\leq \sum_{r=0}^{R-1} 4\eta^2 (1-\beta)^2(\mathcal{E}_{r-1} + \mathbb{E}[\|\nabla f(x^{r-1})\|^2])
$$

$$
+ 2\eta^2\beta^2 \sum_{r=0}^{R-1}\left(\frac{e}{N}\sum_{i=1}^{N}\mathbb{E}[\|\nabla f_i(x^0)\|^2] + 2e(r+1)(\gamma L)^2 \sum_{j=0}^{r-1}(\mathcal{E}_j + \mathbb{E}[\|\nabla f(x^j)\|^2])\right)
$$

$$
\leq 4\eta^2(1-\beta)^2 \sum_{r=0}^{R-1}(\mathcal{E}_{r-1} + \mathbb{E}[\|\nabla f(x^{r-1})\|^2])
$$

$$
+ 2\eta^2\beta^2\left(eRG_0 + 2e(\gamma LR)^2 \sum_{r=0}^{R-2}(\mathcal{E}_r + \mathbb{E}[\|\nabla f(x^r)\|^2])\right).
$$

Rearranging the equation and applying the upper bound of $\eta$ completes the proof. $\qquad\square$

**Theorem 11.** *Under Assumption 1 and 3, if we take $g^0 = 0$,*

$$
\beta = \min\left\{, \sqrt{\frac{NKL\Delta}{\sigma^2 R}}\right\} \text{ for any constant } c \in (0,1], \quad \gamma = \min\left\{\frac{1}{24L}, \frac{\beta}{6L}\right\},
$$

$$
\eta KL \lesssim \min\left\{1, \frac{1}{\beta\gamma LR}, \left(\frac{L\Delta}{G_0\beta^3 R}\right)^{1/2}, \frac{1}{(\beta N)^{1/2}}, \frac{1}{(\beta^3 NK)^{1/4}}\right\}
$$
(5)

*then* FEDAVG-M *converges as*

$$
\frac{1}{R}\sum_{r=0}^{R-1}\mathbb{E}[\|\nabla f(x^r)\|^2] \lesssim \sqrt{\frac{L\Delta\sigma^2}{NKR}} + \frac{L\Delta}{R}.
$$

*Here $G_0 := \frac{1}{N}\sum_{i=1}^{N}\|\nabla f_i(x^0)\|^2$.*

*Proof.* Combining Lemma 8 and 9, we have

$$
\mathcal{E}_r \leq \left(1 - \frac{8\beta}{9}\right)\mathcal{E}_{r-1} + 4\frac{(\gamma L)^2}{\beta}\mathbb{E}[\|\nabla f(x^{r-1})\|^2] + \frac{2\beta^2\sigma^2}{NK}
$$

$$
+ 4\beta L^2\left(2eK^2\Xi_r + K\eta^2\beta^2\sigma^2(1 + 2K^3L^2\eta^2\beta^2)\right).
$$

and

$$
\mathcal{E}_0 \leq (1-\beta)\mathcal{E}_{-1} + \frac{2\beta^2\sigma^2}{NK} + 4\beta L^2\left(2eK^2\Xi_0 + K\eta^2\beta^2\sigma^2(1 + 2K^3L^2\eta^2\beta^2)\right).
$$

Summing over $r$ from 0 to $R-1$ and applying Lemma 10,

$$\sum_{r=0}^{R-1} \mathcal{E}_r \le \left(1 - \frac{8\beta}{9}\right) \sum_{r=-1}^{R-2} \mathcal{E}_r + 4\frac{(\gamma L)^2}{\beta} \sum_{r=0}^{R-2} \mathbb{E}[\|\nabla f(x^r)\|^2] + 2\frac{\beta^2\sigma^2}{NK}R$$

$$+ 4\beta L^2 \left(2eK^2 \sum_{r=0}^{R-1} \Xi_r + RK\eta^2\beta^2\sigma^2(1 + 2K^3L^2\eta^2\beta^2)\right)$$

$$\le \left(1 - \frac{7\beta}{9}\right) \sum_{r=-1}^{R-2} \mathcal{E}_r + \left(4\frac{(\gamma L)^2}{\beta} + \frac{\beta}{9}\right) \sum_{r=-1}^{R-2} \mathbb{E}[\|\nabla f(x^r)\|^2] + 16\beta^3(e\eta KL)^2 RG_0$$

$$+ \frac{2\beta^2\sigma^2}{NK}R + 4\beta^3(\eta KL)^2 \left(\frac{1}{K} + 2(\eta KL\beta)^2\right) \sigma^2 R$$

$$\le \left(1 - \frac{7\beta}{9}\right) \sum_{r=-1}^{R-2} \mathcal{E}_r + \frac{2\beta}{9} \sum_{r=-1}^{R-2} \mathbb{E}[\|\nabla f(x^r)\|^2] + 16\beta^3(e\eta KL)^2 RG_0 + \frac{4\beta^2\sigma^2}{NK}R.$$

Here in the last inequality we apply

$$4\beta(\eta KL)^2 \left(\frac{1}{K} + 2(\eta KL\beta)^2\right) \le \frac{2}{NK} \quad \text{and} \quad \gamma L \le \frac{\beta}{6}.$$

Therefore,

$$\sum_{r=0}^{R-1} \mathcal{E}_r \le \frac{9}{7\beta}\mathcal{E}_{-1} + \frac{2}{7}\mathbb{E}[\sum_{r=-1}^{R-2} \|\nabla f(x^r)\|^2] + \frac{144}{7}(e\beta\eta KL)^2 G_0 R + \frac{36\beta\sigma^2}{7NK}R.$$

Combine this inequality with Lemma 5 and we get

$$\frac{1}{\gamma}\mathbb{E}[f(x^R) - f(x^0)] \le -\frac{1}{7} \sum_{r=0}^{R-1} \mathbb{E}[\|\nabla f(x^r)\|^2] + \frac{39}{56\beta}\mathcal{E}_{-1} + \frac{78}{7}(e\beta\eta KL)^2 G_0 R + \frac{39\beta\sigma^2}{14NK}R.$$

Finally, noticing that $g^0 = 0$ implies $\mathcal{E}_{-1} \le 2L(f(x^0) - f^*) = 2L\Delta$, we obtain

$$\frac{1}{R} \sum_{r=0}^{R-1} \mathbb{E}[\|\nabla f(x^r)\|^2] \lesssim \frac{L\Delta}{\gamma LR} + \frac{\mathcal{E}_{-1}}{\beta R} + (\beta\eta KL)^2 G_0 + \frac{\beta\sigma^2}{NK}$$

$$\lesssim \frac{L\Delta}{R} + \frac{L\Delta}{\beta R} + \frac{\beta\sigma^2}{NK} + (\beta\eta KL)^2 G_0$$

$$\lesssim \frac{L\Delta}{R} + \sqrt{\frac{L\Delta\sigma^2}{NKR}}.$$

$\square$

## B.2 FEDAVG-M-VR

In this subsection, we present the proofs for the FEDAVG-M-VR algorithm, shown as in Algorithm 3.

**Lemma 12.** *If $\gamma L \le \sqrt{\frac{\beta NK}{54}}$, the following holds for $r \ge 1$:*

$$\mathcal{E}_r \le (1 - \frac{8\beta}{9})\mathcal{E}_{r-1} + \frac{4}{\beta}L^2 U_r + \frac{3\beta^2\sigma^2}{NK} + \frac{6(\gamma L)^2}{NK}\mathbb{E}[\|\nabla f(x^{r-1})\|^2].$$

*Also for $r = 0$, it holds that*

$$\mathcal{E}_0 \le (1 - \beta)\mathcal{E}_{-1} + \frac{4}{\beta}L^2 U_r + \frac{3\beta^2\sigma^2}{NK}.$$

---

**Algorithm 3** FEDAVG-M-VR: FEDAVG with variance-reduced momentum

---

**Require:** initial model $x^{-1} = x^0$ and gradient estimate $g^0$, local learning rate $\eta$, global learning rate $\gamma$, momentum $\beta$

  **for** $r = 0, \cdots, R - 1$ **do**

    **for** each client $i \in \{1, \ldots, N\}$ in parallel **do**

      Initial local model $x_i^{r,0} = x^r$

      **for** $k = 0, \cdots, K - 1$ **do**

        Compute direction $g_i^{r,k} = \nabla F(x_i^{r,k}; \xi_i^{r,k}) + (1 - \beta)(g^r - \nabla F(x^{r-1}; \xi_i^{r,k}))$

        Update local model $x_i^{r,k+1} = x_i^{r,k} - \eta g_i^{r,k}$

      **end for**

    **end for**

    Aggregate local updates $g^{r+1} = \frac{1}{\eta N K} \sum_{i=1}^{N} \left( x^r - x_i^{r,K} \right)$

    Update global model global $x^{r+1} = x^r - \gamma g^{r+1}$

  **end for**

---

*Proof.*

$$\mathcal{E}_r = \mathbb{E}\left[\left\| \frac{1}{NK} \sum_{i,\,k} \nabla F(x_i^{r,k}; \xi_i^{r,k}) + (1 - \beta)\left(g^r - \frac{1}{NK} \sum_{i,\,k} \nabla F(x^{r-1}; \xi_i^{r,k})\right) - \nabla f(x^r)\right\|^2\right]$$

$$= \mathbb{E}\left[\left\| (1 - \beta)(g^r - \nabla f(x^{r-1})) + \frac{1}{NK} \sum_{i,\,k} \nabla F(x_i^{r,k}; \xi_i^{r,k}) - \nabla f(x^r) \right.\right.$$

$$\left.\left. + (1 - \beta)\left(\nabla f(x^{r-1}) - \frac{1}{NK} \sum_{i,\,k} \nabla F(x^{r-1}; \xi_i^{r,k})\right) \right\|^2\right]$$

$$= (1 - \beta)^2 \mathcal{E}_{r-1} + \underbrace{2\mathbb{E}\left[\left\langle (1 - \beta)(g^r - \nabla f(x^{r-1})), \frac{1}{NK} \sum_{i,\,k} \nabla f_i(x_i^{r,k}) - \nabla f(x^r) \right\rangle\right]}_{\Lambda_1}$$

$$+ \underbrace{\mathbb{E}\left\| \frac{1}{NK} \sum_{i,\,k} \nabla F(x_i^{r,k}; \xi_i^{r,k}) - \nabla f(x^r) + (1 - \beta)\left(\nabla f(x^{r-1}) - \frac{1}{NK} \sum_{i,\,k} \nabla F(x^{r-1}; \xi_i^{r,k})\right) \right\|^2}_{\Lambda_2}.$$

By the AM-GM inequality and Assumption 2,

$$\Lambda_1 \leq \beta(1 - \beta)^2 \mathcal{E}_{r-1} + \frac{1}{\beta} L^2 U_r.$$

By Assumption 2,

$$\Lambda_2 = \mathbb{E}\left[\left\| \frac{1}{NK} \sum_{i,\,k} (\nabla F(x_i^{r,k}; \xi_i^{r,k}) - \nabla F(x^r; \xi_i^{r,k})) + \beta\left(\frac{1}{NK} \sum_{i,\,k} \nabla F(x_i^{r,k}; \xi_i^{r,k}) - \nabla f(x^r)\right) \right.\right.$$

$$\left.\left. + (1 - \beta)\left(\frac{1}{NK} \sum_{i,\,k} (\nabla F(x^r; \xi_i^{r,k}) - \nabla F(x^{r-1}; \xi_i^{r,k})) - \nabla f(x^r) + \nabla f(x^{r-1})\right) \right\|^2\right]$$

$$\leq 3L^2 U_r + 3\frac{\beta^2 \sigma^2}{NK} + 3(1 - \beta)^2 \frac{L^2}{NK} \mathbb{E}[\|x^r - x^{r-1}\|^2].$$

Therefore, for $r \geq 1$,

$$\mathcal{E}_r \leq (1 - \beta)\mathcal{E}_{r-1} + \frac{4}{\beta} L^2 U_r + \frac{3\beta^2 \sigma^2}{NK} + 3(1 - \beta)^2 \frac{L^2}{NK} \mathbb{E}[\|x^r - x^{r-1}\|^2]$$

$$\leq (1 - \frac{8\beta}{9})\mathcal{E}_{r-1} + \frac{4}{\beta} L^2 U_r + \frac{3\beta^2 \sigma^2}{NK} + \frac{6(\gamma L)^2}{NK} \mathbb{E}[\|\nabla f(x^{r-1})\|^2].$$

The last inequality is derived by $\|x^r - x^{r-1}\|^2 \leq 2\gamma^2(\|\nabla f(x^{r-1})\|^2 + \|g^r - \nabla f(x^{r-1})\|^2)$ and $\gamma L \leq \sqrt{\frac{\beta NK}{54}}$. Similarly, for $r = 0$, we can obtain

$$\mathcal{E}_0 \leq (1-\beta)\mathcal{E}_{-1} + \frac{4}{\beta}L^2 U_0 + \frac{3\beta^2\sigma^2}{NK}.$$

$\square$

**Lemma 13.** *If $\eta K L \leq \frac{1}{4e}$, the following holds:*
$$U_r \leq 4eK^2\Xi_r + 8(\eta K)^2(2(\eta KL)^2 + K^{-1})\left(\beta^2\sigma^2 + 2L^2\mathbb{E}[\|x^r - x^{r-1}\|^2]\right).$$

*Proof.* Note that $\zeta_i^{r,k} = -\eta(\nabla f_i(x_i^{r,k}) + (1-\beta)(g^r - \nabla f_i(x^{r-1})))$. Then we have
$$\mathbb{E}[\|\zeta_i^{r,j} - \zeta_i^{r,j-1}\|^2] \leq \eta^2 L^2 \mathbb{E}[\|x_i^{r,j} - x_i^{r,j-1}\|^2]$$
$$= \eta^2 L^2\left(\mathbb{E}[\|\zeta_i^{r,j-1}\|^2] + \mathbb{E}[\text{Var}[x_i^{r,j} - x_i^{r,j-1}|\mathcal{F}_i^{r,j-1}]]\right).$$
Here we use bias-variance decomposition and $\text{Var}[\cdot|\cdot]$ stands for the conditional variance. Since
$$\mathbb{E}[\text{Var}[x_i^{r,j} - x_i^{r,j-1}|\mathcal{F}_i^{r,j-1}]]$$
$$= \eta^2\mathbb{E}\left[\left\|\nabla F(x_i^{r,j-1};\xi_i^{r,j-1}) - \nabla f_i(x_i^{r,j-1}) - (1-\beta)\left(\nabla F(x^{r-1};\xi_i^{r,j-1}) - \nabla f_i(x^{r-1})\right)\right\|^2\right]$$
$$\leq \eta^2\left(2\beta^2\sigma^2 + 2(1-\beta)^2 L^2\mathbb{E}[\|x^{r-1} - x_i^{r,j-1}\|^2]\right),$$
then
$$\mathbb{E}[\|\zeta_i^{r,j} - \zeta_i^{r,j-1}\|^2]$$
$$\leq \eta^2 L^2\left(\mathbb{E}[\|\zeta_i^{r,j-1}\|^2] + 2\beta^2\eta^2\sigma^2 + 2\eta^2(1-\beta)^2 L^2\mathbb{E}[\|x^{r-1} - x_i^{r,j-1}\|^2]\right)$$
$$\leq \eta^2 L^2\left(\mathbb{E}[\|\zeta_i^{r,j-1}\|^2] + 2\beta^2\eta^2\sigma^2 + 4\eta^2 L^2\mathbb{E}[\|x^{r-1} - x^r\|^2 + \|x^r - x_i^{r,j-1}\|^2]\right).$$
Therefore for any $1 \leq j \leq k-1 \leq K-2$,
$$\mathbb{E}\|\zeta_i^{r,j}\|^2 \leq (1 + \frac{1}{k})\mathbb{E}[\|\zeta_i^{r,j-1}\|^2] + (1+k)\mathbb{E}[\|\zeta_i^{r,j} - \zeta_i^{r,j-1}\|^2]$$
$$\leq \left(1 + \frac{2}{k}\right)\mathbb{E}\|\zeta_i^{r,j-1}\|^2 + (k+1)\eta^2 L^2\left(2\beta^2\eta^2\sigma^2 + 4\eta^2 L^2\mathbb{E}[\|x^{r-1} - x^r\|^2 + \|x^r - x_i^{r,j-1}\|^2]\right)$$
$$\leq e^2\mathbb{E}\|\zeta_i^{r,0}\|^2 + 8k^2 L^2\eta^4(2\beta^2\sigma^2 + 4L^2\mathbb{E}[\|x^r - x^{r-1}\|^2]) + 4e^2 k(\eta L)^4\sum_{j'=0}^{j-1}\mathbb{E}[\|x_i^{r,j'} - x^r\|^2].$$
$$(6)$$
Here the second inequality is by $\eta L \leq \frac{1}{K} \leq \frac{1}{k+1}$. The last inequality is by unrolling the recursive bound and using $\left(1 + \frac{2}{k}\right)^k \leq e^2$. By Lemma 6, it holds that
$$\mathbb{E}[\|x_i^{r,k} - x^r\|^2]$$
$$\leq 2\mathbb{E}\left[\left\|\sum_{j=0}^{k-1}\zeta_i^{r,j}\right\|^2\right] + 2\sum_{j=0}^{k-1}\mathbb{E}[\text{Var}[x_i^{r,j+1} - x_i^{r,j}|\mathcal{F}_i^{r,j}]]$$
$$\leq 2k\sum_{j=0}^{k-1}\mathbb{E}[\|\zeta_i^{r,j}\|^2] + 2\sum_{j=0}^{k-1}\left(2\beta^2\eta^2\sigma^2 + 4\eta^2 L^2\mathbb{E}[\|x^{r-1} - x^r\|^2 + \|x^r - x_i^{r,j}\|^2]\right). \quad (7)$$
Summing up (7) over $k = 0, \ldots, K-1$, using (6) and $8(\eta L)^2 + 8e^2(\eta KL)^4 \leq \frac{1}{2}$ due to the condition on $\eta$, we have
$$\frac{1}{2K}\sum_{k=0}^{K-1}\mathbb{E}[\|x_i^{r,k} - x^r\|^2] \leq 2eK^2\mathbb{E}[\|\zeta_i^{r,0}\|^2] + (8(\eta K)^4 L^2 + 4\eta^2 K)\left(\beta^2\sigma^2 + 2L^2\mathbb{E}[\|x^r - x^{r-1}\|^2]\right).$$
This implies
$$U_r \leq 4eK^2\Xi_r + 8(\eta K)^2(2(\eta KL)^2 + K^{-1})\left(\beta^2\sigma^2 + 2L^2\mathbb{E}[\|x^r - x^{r-1}\|^2]\right).$$

$\square$

**Lemma 14.** *If* $\gamma L \leq \frac{1}{24}$ *and* $288e(\eta KL)^2 \left(\frac{289}{72}(1-\beta)^2 + 8e(\gamma\beta LR)^2\right) \leq \beta^2$, *then the following holds:*

$$\sum_{r=0}^{R-1} \Xi_r \leq \frac{\beta^2}{288eK^2L^2} \sum_{r=-1}^{R-2} (\mathcal{E}_r + \mathbb{E}[\|\nabla f(x^r)\|^2]) + 4\eta^2\beta^2 eRG_0.$$

*Proof.* Recall that $\zeta_i^{r,0} = -\eta((1-\beta)(g^r - \nabla f_i(x^{r-1})) + \nabla f_i(x^r))$. Consequently, we have

$$\|\zeta_i^{r,0}\|^2 \leq 2\eta^2 \left((1-\beta)^2\|g^r\|^2 + \|\nabla f_i(x^r) - (1-\beta)\nabla f_i(x^{r-1})\|^2\right)$$
$$\leq 2\eta^2(1-\beta)^2(1+2(\gamma L)^2)\|g^r\|^2 + 4\eta^2\beta^2\|\nabla f_i(x^r)\|^2$$
$$\leq \frac{289}{144}\eta^2(1-\beta)^2\|g^r\|^2 + 4\eta^2\beta^2\|\nabla f_i(x^r)\|^2.$$

Using Young's inequality, we can obtain that for any $q > 0$,

$$\mathbb{E}[\|\nabla f_i(x^r)\|^2] \leq (1+q)\mathbb{E}[\|\nabla f_i(x^{r-1})\|^2] + (1+q^{-1})L^2\mathbb{E}\|x^r - x^{r-1}\|^2$$
$$\leq (1+q)\mathbb{E}[\|\nabla f_i(x^{r-1})\|^2] + 2(1+q^{-1})(\gamma L)^2(\mathcal{E}_{r-1} + \mathbb{E}[\|\nabla f(x^{r-1})\|^2])$$
$$\leq (1+q)^r\mathbb{E}[\|\nabla f_i(x^0)\|^2] + \frac{2}{q}(\gamma L)^2 \sum_{j=0}^{r-1}(\mathcal{E}_j + \mathbb{E}[\|\nabla f(x^j)\|^2])(1+q)^{r-j}.$$

Taking $q = \frac{1}{r}$ in the above, we have

$$\mathbb{E}[\|\nabla f_i(x^r)\|^2] \leq e\mathbb{E}[\|\nabla f_i(x^0)\|^2] + 2e(r+1)(\gamma L)^2 \sum_{j=0}^{r-1}(\mathcal{E}_j + \mathbb{E}[\|\nabla f(x^j)\|^2]).$$

This inequality holds as well trivially for $r = 0$. Therefore, we have

$$\sum_{r=0}^{R-1} \Xi_r \leq \sum_{r=0}^{R-1} \mathbb{E}\left[\frac{289}{144}\eta^2(1-\beta)^2\|g^r\|^2 + 4\eta^2\beta^2\frac{1}{N}\sum_{i=1}^{N}\|\nabla f_i(x^r)\|^2\right]$$
$$\leq \sum_{r=0}^{R-1} \frac{289}{72}\eta^2(1-\beta)^2(\mathcal{E}_{r-1} + \mathbb{E}[\|\nabla f(x^{r-1})\|^2])$$
$$+ 4\eta^2\beta^2 \sum_{r=0}^{R-1}\left(\frac{e}{N}\sum_i \mathbb{E}[\|\nabla f_i(x^0)\|^2] + 2e(r+1)(\gamma L)^2 \sum_{j=0}^{r-1}(\mathcal{E}_j + \mathbb{E}[\|\nabla f(x^j)\|^2])\right)$$
$$\leq \frac{289}{72}\eta^2(1-\beta)^2 \sum_{r=0}^{R-1}(\mathcal{E}_{r-1} + \mathbb{E}[\|\nabla f(x^{r-1})\|^2])$$
$$4\eta^2\beta^2\left(eRG_0 + 2e(\gamma LR)^2 \sum_{r=0}^{R-2}(\mathcal{E}_r + \mathbb{E}[\|\nabla f(x^r)\|^2])\right)$$
$$\leq \frac{\beta^2}{288eK^2L^2} \sum_{r=-1}^{R-2}(\mathcal{E}_r + \mathbb{E}[\|\nabla f(x^r)\|^2]) + 4\eta^2\beta^2 eRG_0.$$

Here the last inequality is due to the upper bound of $\eta$. $\square$

**Theorem 15.** *Under Assumption 2 and 3, if we take* $g^0 = \frac{1}{NB}\sum_{i=1}^{N}\sum_{b=1}^{B}\nabla F(x^0; \xi_i^b)$ *with* $\{\xi_i^b\}_{b=1}^B \overset{iid}{\sim} \mathcal{D}_i$ *and set*

$$\beta = \min\left\{c, \left(\frac{NKL^2\Delta^2}{\sigma^4 R^2}\right)^{1/3}\right\} \text{ for any constant } c \in (0, 1], \quad \gamma = \min\left\{\frac{1}{24L}, \sqrt{\frac{\beta NK}{54L^2}}\right\},$$

$$\eta KL \lesssim \min\left\{\left(\frac{L\Delta}{G_0\gamma LR}\right)^{1/2}, \left(\frac{\beta}{N}\right)^{1/2}, \left(\frac{\beta}{NK}\right)^{1/4}\right\}, \quad B = \left\lceil\frac{K}{R\beta^2}\right\rceil,$$

$$\tag{8}$$

FEDAVG-M-VR *converges as*

$$\frac{1}{R}\sum_{r=0}^{R-1}\mathbb{E}[\|\nabla f(x^r)\|^2] \lesssim \left(\frac{L\Delta\sigma}{NKR}\right)^{2/3} + \frac{L\Delta}{R}.$$

*Alternatively, if $B = \Theta(KR)$ and $\beta = \min\left\{\frac{1}{R}, \left(\frac{NKL^2\Delta^2}{\sigma^4 R^2}\right)^{1/3}\right\}$, then* FEDAVG-M-VR *converges as*

$$\frac{1}{R}\sum_{r=0}^{R-1}\mathbb{E}[\|\nabla f(x^r)\|^2] \lesssim \left(\frac{L\Delta\sigma}{NKR}\right)^{2/3} + \frac{\sigma^2}{NKR} + \frac{L\Delta}{R}.$$

*Proof.* Combine Lemma 12, 13 and we have

$$\mathcal{E}_r \leq (1-\frac{8\beta}{9})\mathcal{E}_{r-1} + \frac{(6\gamma L)^2}{NK}\mathbb{E}[\|\nabla f(x^{r-1})\|^2] + \frac{3\beta^2\sigma^2}{NK}$$
$$+ \frac{4}{\beta}L^2\left(4eK^2\Xi_r + 8(\eta K)^2(2(\eta KL)^2 + K^{-1})(\beta^2\sigma^2 + 2L^2\mathbb{E}[\|x^r - x^{r-1}\|^2])\right)$$
$$\mathcal{E}_0 \leq (1-\beta)\mathcal{E}_{-1} + \frac{3\beta^2\sigma^2}{NK} + \frac{4}{\beta}L^2\left(4eK^2\Xi_0 + 8(\eta K)^2(2(\eta KL)^2 + K^{-1}))\beta^2\sigma^2\right)$$

Summing over $r$ from 0 to $R-1$ and applying Lemma 14,

$$\sum_{r=0}^{R-1}\mathcal{E}_r$$

$$\leq(1-\frac{8\beta}{9})\sum_{r=-1}^{R-2}\mathcal{E}_r + \frac{6(\gamma L)^2}{NK}\mathbb{E}\left[\sum_{r=0}^{R-2}\|\nabla f(x^r)\|^2\right] + \frac{3\beta^2\sigma^2}{NK}R$$

$$+ \frac{4}{\beta}L^2\left(4eK^2\sum_{r=0}^{R-1}\Xi_r + 8(\eta K)^2(2(\eta KL)^2 + \frac{1}{K})\left(R\beta^2\sigma^2 + 2L^2\sum_{r=0}^{R-1}\mathbb{E}[\|x^r - x^{r-1}\|^2]\right)\right)$$

$$\leq(1-\frac{7\beta}{9})\sum_{r=-1}^{R-2}\mathcal{E}_r + \left(\frac{6(\gamma L)^2}{NK} + \frac{\beta}{9}\right)\mathbb{E}[\sum_{r=-1}^{R-2}\|\nabla f(x^r)\|^2] + 64\beta(e\eta KL)^2 RG_0$$

$$+ \frac{3\beta^2\sigma^2}{NK}R + 32\beta(\eta KL)^2\left(\frac{1}{K} + 2(\eta KL)^2\right)\sigma^2 R$$

$$\leq(1-\frac{7\beta}{9})\sum_{r=-1}^{R-2}\mathcal{E}_r + \frac{2\beta}{9}\mathbb{E}\left[\sum_{r=-1}^{R-2}\|\nabla f(x^r)\|^2\right] + 64\beta(e\eta KL)^2 RG_0 + \frac{4\beta^2\sigma^2}{NK}R.$$

Here in the second inequality, we apply

$$\begin{cases} 32\beta(\eta KL)^2(\frac{1}{K} + 2(\eta KL)^2) \leq \frac{\beta^2}{NK}, \\ \frac{128(\eta KL)^2}{\beta}(\frac{1}{K} + 2(\eta KL)^2)(\gamma L)^2 \leq \frac{\beta}{18}, \\ \gamma L \leq \sqrt{\frac{\beta NK}{54}}. \end{cases}$$

Therefore, we obtain

$$\sum_{r=0}^{R-1}\mathcal{E}_r \leq \frac{9}{7\beta}\mathcal{E}_{-1} + \frac{2}{7}\mathbb{E}\left[\sum_{r=-1}^{R-2}\|\nabla f(x^r)\|^2\right] + \frac{576}{7}(e\eta KL)^2 G_0 R + \frac{36\beta\sigma^2}{7NK}R.$$

Combine this inequality with Lemma 5 and we get

$$\frac{1}{\gamma}\mathbb{E}[f(x^R) - f(x^0)] \leq -\frac{1}{7}\sum_{r=0}^{R-1}\mathbb{E}[\|\nabla f(x^r)\|^2] + \frac{39}{56\beta}\mathcal{E}_{-1} + \frac{312}{7}(e\eta KL)^2 G_0 R + \frac{39\beta\sigma^2}{14NK}R.$$

Finally, for $B = \left\lceil \frac{K}{R\beta^2} \right\rceil$, noticing that $g^0 = \frac{1}{NB} \sum_i \sum_{b=1}^{B} \nabla F(x^0; \xi_i^b)$ implies $\mathcal{E}_{-1} \leq \frac{\sigma^2}{NB} \leq \frac{\beta^2 \sigma^2 R}{NK}$ and thus

$$\frac{1}{R} \sum_{r=0}^{R-1} \mathbb{E}[\|\nabla f(x^r)\|^2] \lesssim \frac{L\Delta}{\gamma LR} + \frac{\mathcal{E}_{-1}}{\beta R} + (\eta KL)^2 G_0 + \frac{\beta \sigma^2}{NK}$$

$$\lesssim \frac{L\Delta}{\gamma LR} + \frac{\beta \sigma^2}{NK}$$

$$\lesssim \frac{L\Delta}{R} + \frac{L\Delta}{\sqrt{\beta NKR}} + \frac{\beta \sigma^2}{NK}$$

$$\lesssim \frac{L\Delta}{R} + \left(\frac{L\Delta \sigma}{NKR}\right)^{2/3}$$

Similarly, for $B = KR$, $\mathcal{E}_{-1} \leq \frac{\sigma^2}{NB} \leq \frac{\sigma^2}{NKR}$, and we have

$$\frac{1}{R} \sum_{r=0}^{R-1} \mathbb{E}[\|\nabla f(x^r)\|^2] \lesssim \frac{L\Delta}{\gamma LR} + \frac{\mathcal{E}_{-1}}{\beta R} + (\eta KL)^2 G_0 + \frac{\beta \sigma^2}{NK}$$

$$\lesssim \frac{L\Delta}{\gamma LR} + \frac{\sigma^2}{\beta NKR^2} + \frac{\beta \sigma^2}{NK}$$

$$\lesssim \frac{L\Delta}{R} + \frac{L\Delta}{\sqrt{\beta NKR}} + \frac{\sigma^2}{\beta NKR^2} + \frac{\beta \sigma^2}{NK}$$

$$\lesssim \frac{L\Delta}{R} + \left(\frac{L\Delta \sigma}{NKR}\right)^{2/3} + \frac{\sigma^2}{NKR}.$$

$\square$

## C SCAFFOLD WITH MOMENTUM

### C.1 SCAFFOLD-M

In this subsection, we present the proofs for the SCAFFOLD-M algorithm.

**Lemma 16.** *If $\gamma L \leq \frac{\beta}{12}$, the following holds for $r \geq 1$:*

$$\mathcal{E}_r \leq \left(1 - \frac{8\beta}{9}\right)\mathcal{E}_{r-1} + \frac{16}{\beta}(\gamma L)^2 \mathbb{E}[\|\nabla f(x^{r-1})\|^2] + \frac{4\beta^2\sigma^2}{SK} + 10\beta L^2 U_r + 6\beta^2 \frac{N-S}{S(N-1)}V_r.$$

*In addition,*

$$\mathcal{E}_0 \leq (1-\beta)\mathcal{E}_{-1} + \frac{4\beta^2\sigma^2}{SK} + 8\beta L^2 U_0 + 4\beta^2 \frac{N-S}{S(N-1)}V_0.$$

*Proof.* Note that $\frac{1}{N}\sum_{i=1}^N c_i^r = c^r$ holds for any $r \geq 0$. Using Lemma 7, we have

$$\mathcal{E}_r = \mathbb{E}\left[\left\|\nabla f(x^r) - \frac{1}{NK}\sum_{i,\,k}g_i^{r,k}\right\|^2\right] + \frac{N-S}{S(N-1)}\frac{1}{N}\sum_{i=1}^N\mathbb{E}\left[\left\|\frac{1}{K}\sum_k g_i^{r,k} - \frac{1}{NK}\sum_{j,k}g_j^{r,k}\right\|^2\right]$$

$$= \underbrace{\mathbb{E}\left[\left\|(1-\beta)(\nabla f(x^r) - g^r) + \beta\left(\frac{1}{NK}\sum_{i,\,k}\nabla F(x_i^{r,k};\xi_i^{r,k}) - \nabla f(x^r)\right)\right\|^2\right]}_{\Lambda_1}$$

$$+ \frac{\beta^2(N-S)}{S(N-1)}\frac{1}{N}\sum_{i=1}^N\underbrace{\mathbb{E}\left[\left\|\frac{1}{K}\sum_k \nabla F(x_i^{r,k};\xi_i^{r,k}) - \frac{1}{NK}\sum_{j,k}\nabla F(x_j^{r,k};\xi_j^{r,k}) - (c_i^r - c^r)\right\|^2\right]}_{\Lambda_2}.$$

For $r \geq 1$, similar to the proof of Lemma 8, we have

$$\Lambda_1 \leq (1-\beta)\mathcal{E}_{r-1} + \frac{2}{\beta}L^2\mathbb{E}[\|x^r - x^{r-1}\|^2] + \frac{2\beta^2\sigma^2}{NK} + 4\beta L^2 U_r.$$

Besides, by AM-GM inequality and Lemma 6,

$$\Lambda_2 \leq \frac{1}{N}\sum_{i=1}^N \mathbb{E}\left[\left\|\frac{1}{K}\sum_k \nabla F(x_i^{r,k};\xi_i^{r,k}) - c_i^r\right\|^2\right]$$

$$\leq \frac{2\sigma^2}{K} + \frac{2}{N}\sum_i \mathbb{E}\left[\left\|\frac{1}{K}\sum_k \nabla f_i(x_i^{r,k}) - c_i^r\right\|^2\right]$$

$$\leq \frac{2\sigma^2}{K} + 6(L^2 U_r + L^2\mathbb{E}[\|x^r - x^{r-1}\|^2] + V_r).$$

Since $\mathbb{E}[\|x^r - x^{r-1}\|^2] \leq 2\gamma^2(\mathcal{E}_{r-1} + \mathbb{E}[\|\nabla f(x^{r-1})\|^2])$ and $\left(\frac{2}{\beta} + 6\beta^2\frac{N-S}{S(N-1)}\right)2(\gamma L)^2 \leq \frac{16}{\beta}(\gamma L)^2 \leq \frac{\beta}{9}$, we have

$$\mathcal{E}_r \leq \left(1 - \frac{8\beta}{9}\right)\mathcal{E}_{r-1} + \frac{16}{\beta}(\gamma L)^2\mathbb{E}[\|\nabla f(x^{r-1})\|^2] + \frac{4\beta^2\sigma^2}{SK} + 10\beta L^2 U_r + 6\beta^2\frac{N-S}{S(N-1)}V_r.$$

The case for $r = 0$ is similar. $\qquad\square$

**Lemma 17.** *If $\gamma L \leq \frac{1}{\sqrt{2\beta}}$ and $\eta KL \leq \frac{1}{\beta}$, it holds for all $r \geq 1$ that*

$$U_r \leq \eta^2 K^2\left(8e(\mathcal{E}_{r-1} + 2\mathbb{E}[\|\nabla f(x^{r-1})\|^2] + \beta^2 V_r) + \beta^2\sigma^2(K^{-1} + 2(\beta\eta KL)^2))\right).$$

*Proof.* Since $\zeta_i^{r,k} = \mathbb{E}[x_i^{r,k+1} - x_i^{r,k}|\mathcal{F}_i^{r,k}] = -\eta(\beta\nabla f_i(x_i^{r,k}) + (1-\beta)g_r - \beta(c_i^r - c^r))$ and $\text{Var}[x_i^{r,k+1} - x_i^{r,k}|\mathcal{F}_i^{r,k}] \leq \beta^2\eta^2\sigma^2$, with exactly the same procedures of Lemma 9, we have

$$U_r \leq 2eK^2\Xi_r + K\eta^2\beta^2\sigma^2(1 + 2K^3L^2\eta^2\beta^2).$$

Additionally, by AM-GM inequality,

$$
\begin{aligned}
\Xi_r &= \frac{\eta^2}{N} \sum_i \mathbb{E}[\|\beta \nabla f_i(x^r) + (1-\beta)g^r - \beta(c_i^r - c^r)\|^2] \\
&= \frac{\eta^2}{N} \sum_i \mathbb{E}\left[\|\beta(\nabla f_i(x^r) - \nabla f_i(x^{r-1})) + (1-\beta)(g^r - \nabla f(x^{r-1}))\right. \\
&\qquad \left. -\beta\left(c_i^r - c^r - \nabla f_i(x^{r-1}) + \nabla f(x^{r-1})\right) + \nabla f(x^{r-1})\|^2\right] \\
&\le 4\eta^2\left(\beta^2 L^2 \mathbb{E}[\|x^r - x^{r-1}\|^2] + (1-\beta)^2 \mathcal{E}_{r-1} + \beta^2 V_r + \mathbb{E}[\|\nabla f(x^{r-1})\|^2]\right) \\
&\le 4\eta^2(\mathcal{E}_{r-1} + 2\mathbb{E}[\|\nabla f(x^{r-1})\|^2] + \beta^2 V_r).
\end{aligned}
$$

Plug this inequality into the above bound completes the proof. $\qquad\square$

**Lemma 18.** *Under the same conditions of Lemma 17, if $\beta\eta KL \le \frac{1}{24K^{1/4}}$ and $\eta K \le \frac{N}{5S}\gamma$, then we have*

$$
\sum_{r=0}^{R-1} V_r \le \frac{3N}{S}\left(V_0 + \frac{4SR}{NK}\sigma^2 + \frac{8N}{S}(\gamma L)^2 \sum_{r=-1}^{R-2}(\mathcal{E}_r + \mathbb{E}[\|\nabla f(x^r)\|^2])\right).
$$

*Proof.* Since

$$
c_i^{r+1} = \begin{cases} c_i^r & \text{with probability } 1 - \frac{S}{N} \\ \frac{1}{K}\sum_k \nabla F(x_i^{r,k}; \xi_i^{r,k}) & \text{with probability } \frac{S}{N}, \end{cases}
$$

using Young's inequality repeatedly, we have

$$
\begin{aligned}
V_{r+1} &= \left(1 - \frac{S}{N}\right)\frac{1}{N}\sum_{i=1}^{N}\mathbb{E}[\|c_i^r - \nabla f_i(x^r)\|^2] + \frac{S}{N}\frac{1}{N}\sum_{i=1}^{N}\mathbb{E}\left[\left\|\frac{1}{K}\sum_k \nabla F(x_i^{r,k};\xi_i^{r,k}) - \nabla f_i(x^r)\right\|^2\right] \\
&\le \left(1 - \frac{S}{N}\right)\frac{1}{N}\sum_{i=1}^{N}\mathbb{E}[\|c_i^r - \nabla f_i(x^r)\|^2] + \frac{S}{N}\left(\frac{2\sigma^2}{K} + 2L^2 U_r\right) \\
&\le \left(1 - \frac{S}{N}\right)\frac{1}{N}\sum_{i=1}^{N}\mathbb{E}\left[\left(1 + \frac{S}{2N}\right)\|c_i^r - \nabla f_i(x^{r-1})\|^2 + \left(1 + \frac{2N}{S}\right)L^2\|x^r - x^{r-1}\|^2\right] \\
&\quad + \frac{2S}{N}\left(\frac{\sigma^2}{K} + L^2 U_r\right) \\
&\le \left(1 - \frac{S}{2N}\right)V_r + \frac{2N}{S}L^2\mathbb{E}[\|x^r - x^{r-1}\|^2] + \frac{2S\sigma^2}{NK} + \frac{2S}{N}L^2 U_r.
\end{aligned}
$$

Here we apply Lemma 6 to obtain the second inequality. Combine this with Lemma 17,

$$
\begin{aligned}
V_{r+1} &\le \left(1 - \frac{S}{2N} + 16e\frac{S}{N}(\beta\eta KL)^2\right)V_r + 2\sigma^2\left(\frac{S}{NK} + \frac{2S}{N}(\beta\eta KL)^2(K^{-1} + 2(\beta\eta KL)^2)\right) \\
&\quad + \left(\frac{4N}{S}(\gamma L)^2 + \frac{32eS}{N}(\eta KL)^2\right)(\mathcal{E}_{r-1} + \mathbb{E}[\|\nabla f(x^{r-1})\|^2]) \\
&\le \left(1 - \frac{S}{3N}\right)V_r + \frac{4S}{NK}\sigma^2 + \frac{8N}{S}(\gamma L)^2(\mathcal{E}_{r-1} + \mathbb{E}[\|\nabla f(x^{r-1})\|^2]),
\end{aligned}
$$

where we apply the upper bound of $\eta$. Therefore, we finish the proof by summing up over $r$ from 0 to $R-1$ and rearranging the inequality. $\qquad\square$

**Theorem 19.** *Under Assumption 1 and 3, if we take $g^0 = 0$, $c_i^0 = \frac{1}{B}\sum_{b=1}^{B}\nabla F(x^0; \xi_i^b)$ with $\{\xi_i^b\}_{b=1}^{B} \overset{iid}{\sim} \mathcal{D}_i$, $c^0 = \frac{1}{N}\sum_{i=1}^{N}c_i^0$ and set*

$$
\gamma = \frac{\beta}{L}, \quad \beta = \min\left\{c, \frac{S}{N^{2/3}}, \sqrt{\frac{L\Delta SK}{\sigma^2 R}}, \sqrt{\frac{L\Delta S^2}{G_0 N}}\right\},
$$

$$
\eta KL \lesssim \min\left\{\frac{1}{S^{1/2}}, \frac{1}{\beta K^{1/4}}, \frac{S^{1/2}}{N}\right\}, \quad B = \left\lceil\frac{NK}{SR}\right\rceil,
$$

(9)

*then* SCAFFOLD-M *converges as*

$$\frac{1}{R}\sum_{r=0}^{R-1}\mathbb{E}[\|\nabla f(x^r)\|^2] \lesssim \sqrt{\frac{L\Delta\sigma^2}{SKR}} + \frac{L\Delta}{R}\left(1 + \frac{N^{2/3}}{S}\right).$$

*Proof.* By Lemma 16, sum over $r$ from 0 to $R-1$ and plug Lemma 17, Lemma 18 in,

$$\sum_{r=0}^{R-1}\mathcal{E}_r \le \left(1 - \frac{8\beta}{9}\right)\sum_{r=-1}^{R-2}\mathcal{E}_r + \frac{16}{\beta}(\gamma L)^2 \sum_{r=0}^{R-2}\mathbb{E}[\|\nabla f(x^r)\|^2]$$

$$+ \frac{4\beta^2\sigma^2}{SK}R + 10\beta L^2\sum_{r=0}^{R-1}U_r + 6\beta^2\frac{N-S}{S(N-1)}\sum_{r=0}^{R-1}V_r$$

$$\le \left(1 - \frac{8\beta}{9} + 80e\beta(\eta KL)^2\right)\sum_{r=-1}^{R-2}\mathcal{E}_r + \left(\frac{16}{\beta}(\gamma L)^2 + 160e\beta(\eta KL)^2\right)\sum_{r=0}^{R-2}\mathbb{E}[\|\nabla f(x^r)\|^2]$$

$$+ \beta^2\sigma^2 R\left(\frac{4}{SK} + 10(\eta KL)^2(K^{-1} + 2(\beta\eta KL)^2)\right) +$$

$$+ \beta^2\left(6\frac{N-S}{S(N-1)} + 80e\beta(\eta KL)^2\right)\sum_{r=0}^{R-1}V_r$$

$$\le \left(1 - \frac{7\beta}{9}\right)\sum_{r=-1}^{R-2}\mathcal{E}_r + \left(\frac{16}{\beta}(\gamma L)^2 + \frac{\beta}{9}\right)\sum_{r=0}^{R-2}\mathbb{E}[\|\nabla f(x^r)\|^2] + \frac{80\beta^2\sigma^2}{SK}R + \frac{30\beta^2 N}{S^2}V_0.$$

Here the coefficients in the last inequality are derived by the following bounds:

$$\begin{cases} 160e\beta(\eta KL)^2 + 24(\frac{\beta\gamma LN}{S})^2\left(6\frac{N-S}{S(N-1)} + 80e\beta(\eta KL)^2\right) \le \frac{\beta}{9}, \\ \\ 10(\eta KL)^2(K^{-1} + 2(\beta\eta KL)^2) + 960e\beta K^{-1}(\eta KL)^2 \le \frac{4}{SK}, \\ \\ 80e\beta(\eta KL)^2 \le \frac{4}{S}, \end{cases}$$

which can be guaranteed by

$$\begin{cases} \gamma L \lesssim \frac{S^{3/2}}{\beta^{1/2}N}, \\ \\ \eta KL \lesssim \frac{1}{S^{1/2}}. \end{cases}$$

Therefore,

$$\sum_{r=0}^{R-1}\mathcal{E}_r \le \frac{9}{7\beta}\mathcal{E}_{-1} + \frac{2}{7}\mathbb{E}\left[\sum_{r=-1}^{R-2}\|\nabla f(x^r)\|^2\right] + \frac{270\beta N}{7S^2}V_0 + \frac{720\beta\sigma^2}{7SK}R.$$

Combining this inequality with Lemma 5, we obtain

$$\frac{1}{\gamma}\mathbb{E}[f(x^R) - f(x^0)] \le -\frac{1}{7}\sum_{r=0}^{R-1}\mathbb{E}[\|\nabla f(x^r)\|^2] + \frac{39}{56\beta}\mathcal{E}_{-1} + \frac{585\beta N}{28S^2}V_0 + \frac{390\beta\sigma^2}{7SK}R.$$

Finally, noticing that $g^0 = 0$ implies $\mathcal{E}_{-1} \le 2L\Delta$ and $c_i = \frac{1}{B}\sum_b \nabla F(x^0; \xi_i^b)$ implies $V_0 \le \frac{\sigma^2}{B} \le \frac{SR\sigma^2}{NK}$, we reach

$$\frac{1}{R}\sum_{r=0}^{R-1}\mathbb{E}[\|\nabla f(x^r)\|^2] \lesssim \frac{L\Delta}{\gamma LR} + \frac{\mathcal{E}_{-1}}{\beta R} + \frac{\beta N}{S^2 R}V_0 + \frac{\beta\sigma^2}{SK}$$

$$\lesssim \frac{L\Delta}{\beta R} + \frac{L\Delta}{S^{3/2}R}N\beta^{1/2} + \frac{\beta\sigma^2}{SK}$$

$$\lesssim \frac{L\Delta}{R}\left(1 + \frac{N^{2/3}}{S}\right) + \sqrt{\frac{L\Delta\sigma^2}{SKR}}.$$

$\square$

## C.2 SCAFFOLD-M-VR

In this subsection, we present the proofs for the SCAFFOLD-M-VR algorithm, shown as in Algorithm 4.

---

**Algorithm 4** SCAFFOLD-M-VR: SCAFFOLD with variance-reduced momentum

---

**Require:** initial model $x^{-1} = x_0$, gradient estimator $g^0$, control variables $\{c_i^0\}_{i=1}^N$ and $c^0$, local learning rate $\eta$, global learning rate $\gamma$, momentum $\beta$
  **for** $r = 0, \cdots, R - 1$ **do**
    Uniformly sample clients $\mathcal{S}_r \subseteq \{1, \cdots, N\}$ with $|\mathcal{S}_r| = S$
    **for** each client $i \in \mathcal{S}_r$ in parallel **do**
      Initialize local model $x_i^{r,0} = x^r$
      **for** $k = 0, \cdots, K - 1$ **do**
        Compute $g_i^{r,k} = \nabla F(x_i^{r,k}; \xi_i^{r,k}) - \beta(c_i^r - c^r) + (1 - \beta)(g^r - \nabla F(x^{r-1}; \xi_i^{r,k}))$
        Update local model $x_i^{r,k+1} = x_i^{r,k} - \eta g_i^{r,k}$
      **end for**
      Update control variable $c_i^{r+1} := \frac{1}{K} \sum_k \nabla F(x_i^{r,k}; \xi_i^{r,k})$ (for $i \notin \mathcal{S}_r$, $c_i^{r+1} = c_i^r$)
    **end for**
    Aggregate local updates $g^{r+1} = \frac{1}{\eta S K} \sum_{i \in \mathcal{S}_r} \left( x^r - x_i^{r,K} \right)$
    Update global model $x^{r+1} = x^r - \gamma g^{r+1}$
    Update control variable $c^{r+1} = c^r + \frac{1}{N} \sum_{i \in \mathcal{S}_r} (c_i^{r+1} - c_i^r)$
  **end for**

---

**Lemma 20.** *If $\gamma L \leq \sqrt{\frac{\beta S}{126}}$, then the following holds for $r \geq 1$:*

$$\mathcal{E}_r \leq (1 - \frac{8\beta}{9})\mathcal{E}_{r-1} + \frac{14(\gamma L)^2}{S}\mathbb{E}[\|\nabla f(x^{r-1})\|^2] + \frac{8}{\beta}L^2 U_r + \frac{7\beta^2 \sigma^2}{SK} + \frac{4(N - S)}{S(N - 1)}\beta^2 V_r.$$

*In addition,*

$$\mathcal{E}_0 \leq (1 - \beta)\mathcal{E}_{-1} + \frac{8}{\beta}L^2 U_0 + \frac{7\beta^2 \sigma^2}{SK} + \frac{4(N - S)}{S(N - 1)}\beta^2 V_0.$$

*Proof.* By Lemma 6, we have

$$\mathcal{E}_r \leq \mathbb{E}\left[\underbrace{\left\|\nabla f(x^r) - \frac{1}{NK}\sum_{i,\,k}\left[\nabla F(x_i^{r,k}; \xi_i^{r,k}) + (1 - \beta)(g^r - \nabla F(x^{r-1}; \xi_i^{r,k}))\right]\right\|^2}_{\Lambda_1}\right]$$

$$+ \underbrace{\frac{N - S}{S(N - 1)}\frac{1}{N}\sum_{i=1}^N \mathbb{E}\left[\left\|\frac{1}{K}\sum_k\left[\nabla F(x_i^{r,k}; \xi_i^{r,k}) - (1 - \beta)\nabla F(x^{r-1}; \xi_i^{r,k})\right] - \beta c_i^r\right\|^2\right]}_{\Lambda_2}.$$

Applying the same derivation as Lemma 12, we can show that

$$\Lambda_1 \leq (1 - \beta)\mathcal{E}_{r-1} + \frac{4}{\beta}L^2 U_r + 3\frac{\beta^2 \sigma^2}{NK} + 3(1 - \beta)^2 \frac{L^2}{NK}\mathbb{E}[\|x^r - x^{r-1}\|^2].$$

Additionally, by the AM-GM inequality,

$$
\Lambda_2 \leq \frac{1}{N}\sum_{i=1}^{N} 4\mathbb{E}\left[\left\|\frac{1}{K}\sum_k \nabla F(x_i^{r,k};\xi_i^{r,k}) - \nabla F(x^r;\xi_i^{r,k})\right\|^2\right.
$$

$$
+ \beta^2 \left\|\frac{1}{K}\sum_k \nabla F(x^r;\xi_i^{r,k}) - \nabla f_i(x^r)\right\|^2 + \beta^2\|\nabla f_i(x^{r-1}) - c_i^r\|^2
$$

$$
+ \left.\left\|\beta(\nabla f_i(x^r) - \nabla f_i(x^{r-1})) + \frac{1-\beta}{K}\sum_k \nabla F(x^r;\xi_i^{r,k}) - \nabla F(x^{r-1};\xi_i^{r,k})\right\|^2\right]
$$

$$
\leq 4\left(L^2 U_r + \frac{\beta^2\sigma^2}{K} + \beta^2 V_r + L^2\mathbb{E}[\|x^r - x^{r-1}\|^2]\right).
$$

Further notice that for $r \geq 1$, $\mathbb{E}[\|x^r - x^{r-1}\|^2] \leq 2\gamma^2(\mathcal{E}_{r-1} + \mathbb{E}[\|\nabla f(x^{r-1})\|^2])$ and

$$
(\gamma L)^2\left(\frac{8(N-S)}{S(N-1)} + \frac{6(1-\beta)^2}{NK}\right) \leq \frac{14(\gamma L)^2}{S} \leq \frac{\beta}{9}.
$$

Hence we obtain

$$
\mathcal{E}_r \leq (1 - \frac{8\beta}{9})\mathcal{E}_{r-1} + \frac{14(\gamma L)^2}{S}\mathbb{E}[\|\nabla f(x^{r-1})\|^2] + \frac{8}{\beta}L^2 U_r + \frac{7\beta^2\sigma^2}{SK} + \frac{4(N-S)}{S(N-1)}\beta^2 V_r.
$$

The case for $r = 0$ can be established similarly. □

**Lemma 21.** *If $\eta KL \leq \frac{1}{4e}$, $\eta K \leq \frac{\gamma N}{10S}$, and $\gamma L \leq \frac{1}{24}$, then it holds that*

$$
\sum_{r=0}^{R-1} V_r \leq \frac{3N}{S}\left(V_0 + \frac{4SR}{NK}\sigma^2 + \frac{6N}{S}(\gamma L)^2\sum_{r=-1}^{R-2}(\mathcal{E}_r + \mathbb{E}[\|\nabla f(x^r)\|^2])\right).
$$

*Proof.* Note that $\zeta_i^{r,k} = -\eta(\nabla f_i(x_i^{r,k}) + (1-\beta)(g^r - \nabla f_i(x^{r-1})) - \beta(c_i^r - c^r))$, with the same procedures in Lemma 13, we have

$$
U_r \leq 4eK^2\Xi_r + 8(\eta K)^2(2(\eta KL)^2 + K^{-1})\left(\beta^2\sigma^2 + 2L^2\mathbb{E}[\|x^r - x^{r-1}\|^2]\right).
$$

Additionally, by the AM-GM inequality,

$$
\Xi_r = \frac{\eta^2}{N}\sum_i \mathbb{E}[\|\nabla f_i(x^r) + (1-\beta)(g^r - \nabla f_i(x^{r-1})) - \beta(c_i^r - c^r)\|^2]
$$

$$
= \frac{\eta^2}{N}\sum_i \mathbb{E}\left[\left\|(\nabla f_i(x^r) - \nabla f_i(x^{r-1})) + (1-\beta)(g^r - \nabla f(x^{r-1}))\right.\right.
$$

$$
\left.\left. - \beta\left(c_i^r - c^r - \nabla f_i(x^{r-1}) + \nabla f(x^{r-1})\right) + \nabla f(x^{r-1})\right\|^2\right]
$$

$$
\leq 4\eta^2\mathbb{E}\left[L^2\|x^r - x^{r-1}\|^2 + (1-\beta)^2\mathcal{E}_{r-1} + \beta^2 V_r + \|\nabla f(x^{r-1})\|^2\right]
$$

$$
\leq 8\eta^2(\mathcal{E}_{r-1} + \mathbb{E}[\|\nabla f(x^{r-1})\|^2] + \beta^2 V_r).
$$

Hence, by applying $32(2(\eta KL)^2 + K^{-1})(\gamma L)^2 \leq 96(\gamma L)^2 \leq 2$, we obtain

$$
U_r \leq 32e(\eta K)^2(\mathcal{E}_{r-1} + \mathbb{E}[\|\nabla f(x^{r-1})\|^2] + \beta^2 V_r)
$$

$$
+ 8(\eta K)^2(2(\eta KL)^2 + K^{-1})\left(\beta^2\sigma^2 + 2L^2\mathbb{E}[\|x^r - x^{r-1}\|^2]\right) \tag{10}
$$

$$
\leq 90(\eta K)^2(\mathcal{E}_{r-1} + \mathbb{E}[\|\nabla f(x^{r-1})\|^2] + \beta^2 V_r) + 8(\beta\eta K)^2(2(\eta KL)^2 + K^{-1})\sigma^2.
$$

Also, similar to Lemma 18, it still holds that

$$
V_{r+1} \leq \left(1 - \frac{S}{2N}\right)V_r + \frac{2N}{S}L^2\mathbb{E}[\|x^r - x^{r-1}\|^2] + \frac{2S\sigma^2}{NK} + \frac{2S}{N}L^2 U_r.
$$

Combine this with the upper bound of $U_r$,

$$V_{r+1}$$

$$\leq \left(1 - \frac{S}{2N} + \frac{180(\beta\eta KL)^2 S}{N}\right) V_r + \left(\frac{4N(\gamma L)^2}{S} + \frac{180(\eta KL)^2 S}{N}\right)(\mathcal{E}_{r-1} + \mathbb{E}[\|\nabla f(x^{r-1})\|^2])$$

$$+ \sigma^2 \left(\frac{2S}{NK} + 8(\beta\eta KL)^2(2(\eta KL)^2 + K^{-1})\right)$$

$$\leq \left(1 - \frac{S}{3N}\right) V_r + \frac{6N(\gamma L)^2}{S}(\mathcal{E}_{r-1} + \mathbb{E}[\|\nabla f(x^{r-1})\|^2]) + \frac{4S\sigma^2}{NK},$$

where we apply the upper bound of $\eta$ in the last inequality. Iterating the above inequality completes the proof. $\qquad\square$

**Theorem 22.** *Under Assumption 2 and 3, if we take $c_i^0 = \frac{1}{B}\sum_{b=1}^{B}\nabla F(x^0;\xi_i^b)$ with $\{\xi_i^b\}_{b=1}^{B} \overset{iid}{\sim} \mathcal{D}_i$, $g^0 = c^0 = \frac{1}{N}\sum_{i=1}^{N}c_i^0$ and set*

$$\gamma = \min\left\{\frac{1}{L}, \frac{\sqrt{\beta S}}{L}\right\}, \quad \beta = \min\left\{\frac{S}{N}, \left(\frac{KL\Delta}{\sigma^2 R}\right)^{2/3} S^{1/3}\right\},$$

$$\eta KL \lesssim \min\left\{\left(\frac{\beta}{S}\right)^{1/2}, \left(\frac{\beta}{SK}\right)^{1/4}\right\}, \quad B = \left\lceil\max\left\{\frac{SK}{NR\beta^2}, \frac{NK}{SR}\right\}\right\rceil,$$

(11)

*SCAFFOLD-M-VR converges as*

$$\frac{1}{R}\sum_{r=0}^{R-1}\mathbb{E}[\|\nabla f(x^r)\|^2] \lesssim \left(\frac{L\Delta\sigma}{S\sqrt{KR}}\right)^{2/3} + \frac{L\Delta}{R}\left(1 + \frac{N^{1/2}}{S}\right).$$

*Alternatively, if $R \gtrsim \frac{N}{S}$ and $\beta = \min\left\{\frac{1}{R}, \left(\frac{KL\Delta}{\sigma^2 R}\right)^{2/3} S^{1/3}\right\}$, $B = \Theta(\frac{SKR}{N})$, SCAFFOLD-M-VR converges as*

$$\frac{1}{R}\sum_{r=0}^{R-1}\mathbb{E}[\|\nabla f(x^r)\|^2] \lesssim \left(\frac{L\Delta\sigma}{S\sqrt{KR}}\right)^{2/3} + \frac{L\Delta}{R}\left(1 + \frac{N^{1/2}}{S} + \frac{\sigma^2}{SKR}\right).$$

*Proof.* By Lemma 20, sum over $r$ from 0 to $R-1$ and plug (10), Lemma 21 in,

$$\sum_{r=0}^{R-1}\mathcal{E}_r \leq (1 - \frac{8\beta}{9})\sum_{r=-1}^{R-2}\mathcal{E}_r + \frac{14(\gamma L)^2}{S}\sum_{r=0}^{R-2}\mathbb{E}[\|\nabla f(x^r)\|^2] + \frac{7\beta^2\sigma^2}{SK}R$$

$$+ \frac{8}{\beta}L^2\sum_{r=0}^{R-1}U_r + 4\beta^2\frac{N-S}{S(N-1)}\sum_{r=0}^{R-1}V_r$$

$$\leq (1 - \frac{8\beta}{9} + 720\frac{(\eta KL)^2}{\beta})\sum_{r=-1}^{R-2}\mathcal{E}_r + (\frac{14(\gamma L)^2}{S} + 720\frac{(\eta KL)^2}{\beta})\sum_{r=0}^{R-2}\mathbb{E}[\|\nabla f(x^r)\|^2]$$

$$+ \beta^2\sigma^2 R\left(\frac{7}{SK} + \frac{64(\eta KL)^2}{\beta}(K^{-1} + 2(\eta KL)^2)\right)$$

$$+ \beta^2\left(\frac{4(N-S)}{S(N-1)} + 720\frac{(\eta KL)^2}{\beta}\right)\sum_{r=0}^{R-1}V_r$$

$$\leq (1 - \frac{7\beta}{9})\sum_{r=-1}^{R-2}\mathcal{E}_r + (\frac{14(\gamma L)^2}{S} + \frac{\beta}{9})\sum_{r=0}^{R-2}\mathbb{E}[\|\nabla f(x^r)\|^2] + 60\frac{\beta^2\sigma^2}{SK}R + 15\frac{\beta^2 N}{S^2}V_0.$$

Here the coefficients in the last inequality are derived by the following bounds:

$$\begin{cases} 720\frac{(\eta KL)^2}{\beta} + 18(\frac{\beta\gamma LN}{S})^2\left(4\frac{N-S}{S(N-1)} + 720\frac{(\eta KL)^2}{\beta}\right) \leq \frac{\beta}{9}, \\[2mm] 64\frac{(\eta KL)^2}{\beta}(K^{-1} + 2(\eta KL)^2) + 8640\frac{(\eta KL)^2}{\beta K} \leq \frac{5}{SK}, \\[2mm] 720\frac{(\eta KL)^2}{\beta} \leq \frac{1}{S}, \end{cases}$$

which can be guaranteed by

$$\begin{cases} \gamma L \lesssim \frac{S^{3/2}}{\beta^{1/2}N}, \\[2mm] \eta KL \lesssim \min\{\sqrt{\frac{\beta}{S}}, (\frac{\beta}{SK})^{1/4}\}. \end{cases}$$

Therefore, it holds that

$$\sum_{r=0}^{R-1}\mathcal{E}_r \leq \frac{9}{7\beta}\mathcal{E}_{-1} + \frac{2}{7}\mathbb{E}\left[\sum_{r=-1}^{R-2}\|\nabla f(x^r)\|^2\right] + \frac{135\beta N}{7S^2}V_0 + \frac{540\beta\sigma^2}{7SK}R.$$

Combine this inequality with Lemma 5 and we get

$$\frac{1}{\gamma}\mathbb{E}[f(x^R) - f(x^0)] \leq -\frac{1}{7}\sum_{r=0}^{R-1}\mathbb{E}[\|\nabla f(x^r)\|^2] + \frac{39}{56\beta}\mathcal{E}_{-1} + \frac{585\beta N}{56S^2}V_0 + \frac{585\beta\sigma^2}{14SK}R.$$

Finally, for $B = \left\lceil \max\left\{\frac{SK}{NR\beta^2}, \frac{NK}{SR}\right\}\right\rceil$, noticing that $g^0 = \frac{1}{NB}\sum_{i,b}\nabla F(x^0;\xi_i^b)$ implies $\mathcal{E}_{-1} \leq \frac{\sigma^2}{NB} \leq \frac{\beta^2\sigma^2 R}{SK}$ and $c_i = \frac{1}{B}\sum_b\nabla F(x^0;\xi_i^b)$ implies $V_0 \leq \frac{\sigma^2}{B} \leq \frac{SR\sigma^2}{NK}$, we reach

$$\frac{1}{R}\sum_{r=0}^{R-1}\mathbb{E}[\|\nabla f(x^r)\|^2] \lesssim \frac{L\Delta}{\gamma LR} + \frac{\mathcal{E}_{-1}}{\beta R} + \frac{\beta N}{S^2 R}V_0 + \frac{\beta\sigma^2}{SK}$$

$$\lesssim \frac{L\Delta}{R} + \frac{L\Delta}{(\beta S)^{1/2}R} + \frac{L\Delta}{S^{3/2}R}N\beta^{1/2} + \frac{\beta\sigma^2}{SK}$$

$$\lesssim \frac{L\Delta}{R}\left(1 + \frac{N^{1/2}}{S}\right) + \left(\frac{L\Delta\sigma}{S\sqrt{KR}}\right)^{2/3}.$$

Similarly, for $B = \frac{SKR}{N}$ and $R \gtrsim \frac{N}{S}$, $\mathcal{E}_{-1} \leq \frac{\sigma^2}{NB} \leq \frac{\sigma^2}{SKR}$, $V_0 \leq \frac{\sigma^2}{B} \leq \frac{N\sigma^2}{SKR}$ and thus we have

$$\frac{1}{R}\sum_{r=0}^{R-1}\mathbb{E}[\|\nabla f(x^r)\|^2] \lesssim \frac{L\Delta}{\gamma LR} + \frac{\mathcal{E}_{-1}}{\beta R} + \frac{\beta N}{S^2 R}V_0 + \frac{\beta\sigma^2}{SK}$$

$$\lesssim \frac{L\Delta}{R} + \frac{L\Delta}{(\beta S)^{1/2}R} + \frac{L\Delta}{S^{3/2}R}N\beta^{1/2} + \frac{\sigma^2}{\beta SKR^2} + \frac{\beta\sigma^2}{SK}$$

$$\lesssim \frac{L\Delta}{R}\left(1 + \frac{N^{1/2}}{S}\right) + \left(\frac{L\Delta\sigma}{S\sqrt{KR}}\right)^{2/3} + \frac{\sigma^2}{SKR}.$$

$\square$

# D IMPLEMENTATION DETAILS & MORE EXPERIMENTS

## D.1 TRAINING SETUP OF MLP

We generate non-iid data for the clients, we sample label ratios from the Dirichlet distribution (Hsu et al., 2019) with a parameter of $0.5$ for the full participation setting and $0.2$ for the partial participation setting. Our experimental setup involves $N = 10$ clients and $K = 32$ local updates. The weight decay is set as $10^{-4}$. The global learning rate is fixed as $\gamma = \eta K$ for all the algorithms, and we perform a grid search for the local learning rate $\eta$ in values $\{0.005, 0.01, 0.05, 0.1, 0.5\}$. Similarly, we search for the momentum parameter $\beta$ in values $\{0.1, 0.2, 0.5, 0.8\}$.

## D.2 TRAINING SETUP OF RESNET18

We generate non-iid data by setting the parameter of Dirichlet distribution as $0.1$, which implies higher heterogeneity. The experiment involves $N = 10$ clients and $K = 16$ local updates. We set $S = 2$ in the partial participation setting. The local learning is fixed as $\hat{\eta} = 0.001$ and global learning rate is $\hat{\gamma} = \hat{\eta} K$. The momentum parameter is $\beta = 0.1$ and batchsize is $128$.

**Reparameterizing momentum.** The update rule of FEDAVG-M in (1) is equivalent to, with a transformation of hyperparameters $\hat{g}_i^{r,k} := g_i^{r,k}/\beta$, $\hat{g}^r := g^r/\beta$, $\hat{\eta} := \beta\eta$, $\hat{\gamma} := \beta\gamma$,

$$\hat{g}_i^{r,k} = \nabla F(x_i^{r,k}; \xi_i^{r,k}) + (1 - \beta)\hat{g}^r,$$

$$x_i^{r,k+1} = x_i^{r,k} - \hat{\eta}\, \hat{g}_i^{r,k}, \quad x^{r+1} = x^r - \hat{\gamma}\hat{g}^{r+1}.$$

This is typically used in the current Pytorch implementation of momentum-based methods. When $\beta = 1$, it still reduces to vanilla FEDAVG. Similarly, in SCAFFOLD-M, the update rule (3) is equivalent to

$$\hat{g}_i^{r,k} = (\nabla F(x_i^{r,k}; \xi_i^{r,k}) - c_i^r + c^r) + (1 - \beta)\hat{g}^r$$

In all the experiments on ResNet18, we implement our proposed FEDAVG-M and SCAFFOLD-M with this reparameterization.

## D.3 MORE EXPERIMENTS OF RESNET18

We conduct more algorithms under mild heterogeneity with the parameter of Dirichlet distribution being $0.5$. We set $S = 5$ in the partial participation setting. Other hyperparameters are the same as described in Section D.2. We plot the evolution of test loss in Figures 3(a) and 3(b), respectively. Again, we observe that our proposed FEDAVG-M and SCAFFOLD-M outperform the vanilla FEDAVG and SCAFFOLD with evident margins. We also evaluate VR methods in terms of test accuracy in the context of full client participation. The results are presented in Figure 4, which demonstrates the advantage of our proposed VR methods over the prior methods.

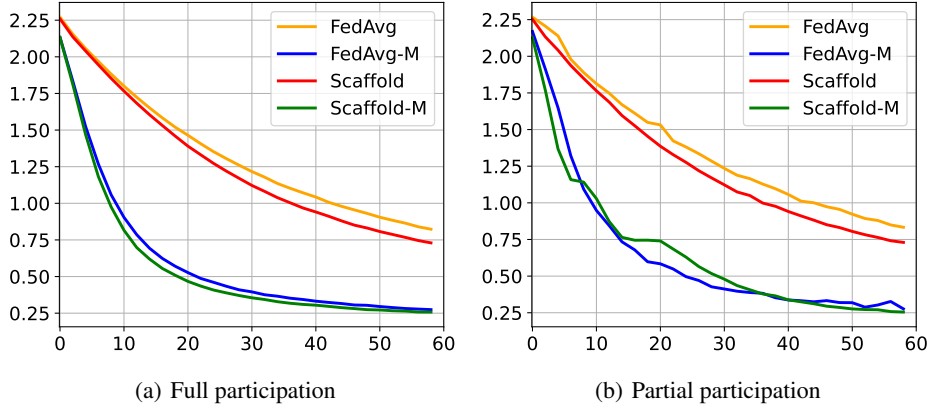

(a) Full participation  (b) Partial participation

Figure 3: Test loss of ResNet18 versus the number of communication rounds

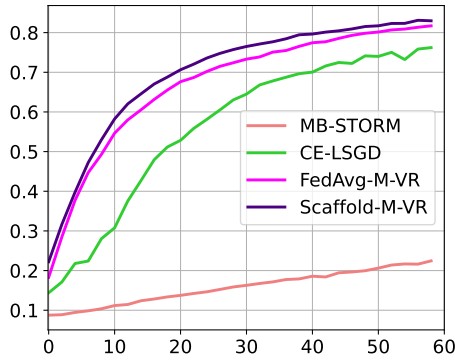

Figure 4: Comparing the test accuracy of VR methods with ResNet-18

## D.4  EXPERIMENTS WITH MORE CLIENTS

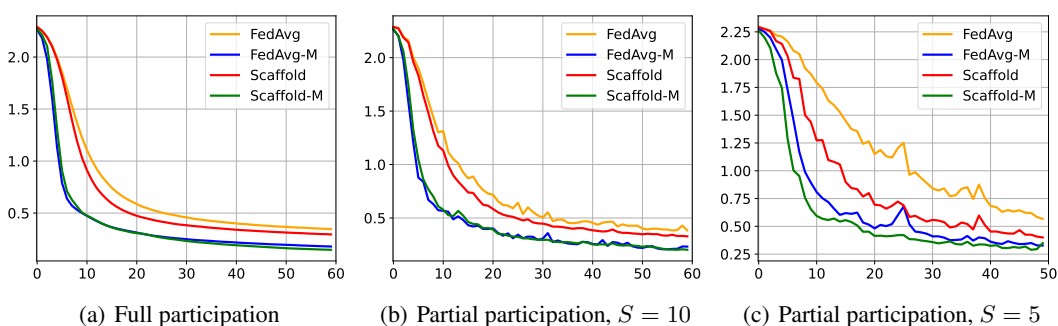

(a) Full participation    (b) Partial participation, $S = 10$    (c) Partial participation, $S = 5$

Figure 5: Test loss of MNIST versus the number of communication rounds

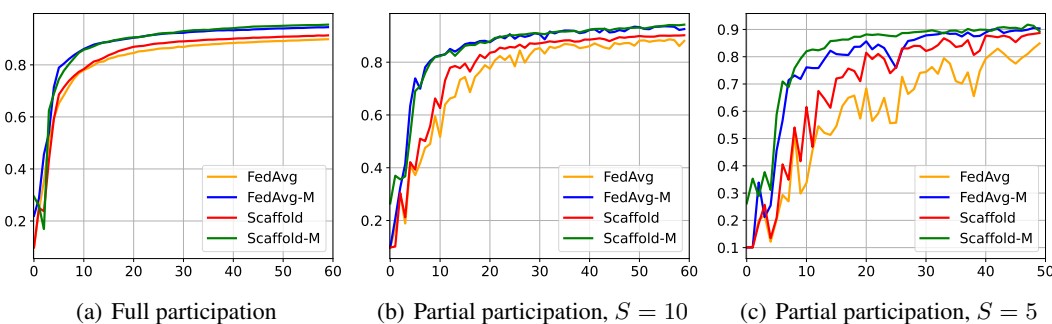

(a) Full participation    (b) Partial participation, $S = 10$    (c) Partial participation, $S = 5$

Figure 6: Test accuracy of MNIST versus the number of communication rounds

We further conduct experiments with $N = 100$ on the MNIST dataset and two-layer fully connected ReLU neural network. The parameter of Dirichlet distribution is $0.2$ and the batchsize is $32$. We set the number of local steps $K = 16$. For FEDAVG-M and SCAFFOLD-M, we set $\beta = 0.2$. We plot the test loss and test accuracy of our proposed algorithms in the regime of full participation ($S = N$) and partial participation ($S = 10$ and $S = 5$). The results are shown in Figure 5 and 6. Compared to former experiments where $N$, we observe that our proposed momentum-based algorithms scale well to FL setups with large $N$. Moreover, we observe that the advantage of our momentum-based variants over the vanilla FEDAVG and SCAFFOLD becomes more evident when fewer clients participate in training, suggesting a great utility of our algorithms in practical FL setups.

### D.5    IMPACT OF MOMENTUM VALUE $\beta$

To further illustrate the effect of momentum, we examine different choices of $\beta$ in both FEDAVG-M and SCAFFOLD-M under partial participation setting with $S = 5$ and $N = 100$. We again simulate with the MNIST dataset and two-layer fully connected ReLU neural networks. The results are shown in Figure 7 and 8. It is worth noting that when $\beta \to 1$, the momentum will anneal down to off, recovering the vanilla FEDAVG and SCAFFOLD. We observe that the stronger the momentum used, the better performance we eventually obtain. This directly demonstrates the benefit of momentum.

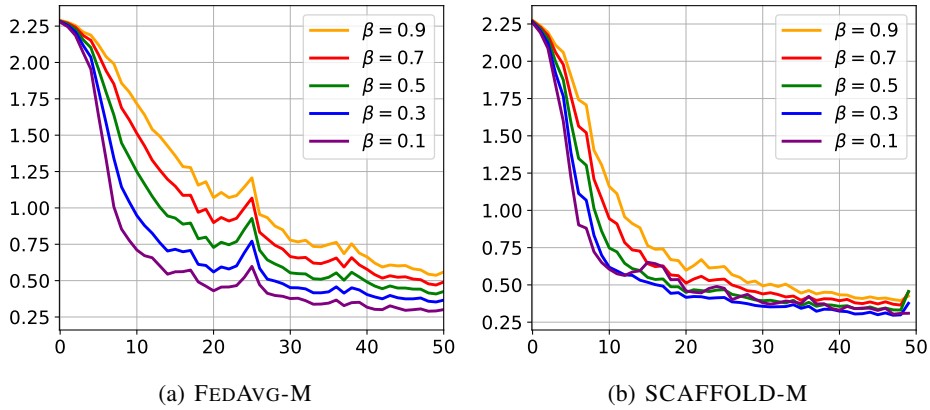

(a) FEDAVG-M                    (b) SCAFFOLD-M

Figure 7: Test loss versus communication rounds with different momentum values

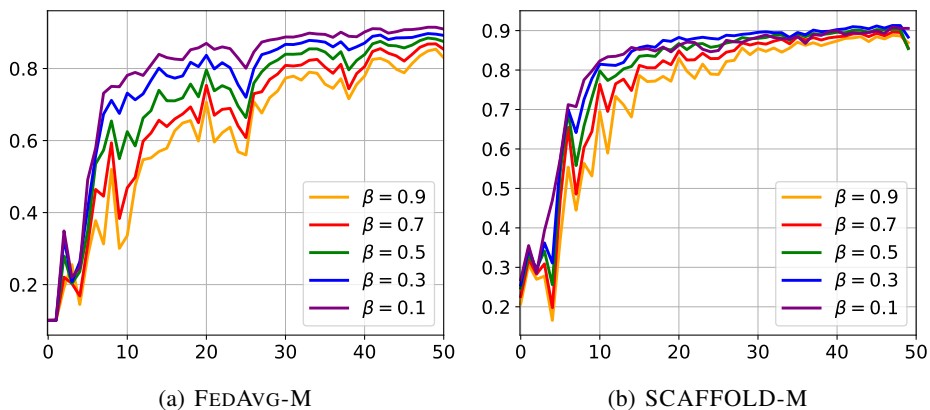

(a) FEDAVG-M                    (b) SCAFFOLD-M

Figure 8: Test accuracy versus communication rounds with different momentum values

