# OpenReview forum: "Momentum Benefits Non-iid Federated Learning Simply and Provably"
_ICLR.cc/2024/Conference — ICLR 2024 poster_

### Official Review · Reviewer_W4SM · 2023-10-28

**Soundness:** 2 fair
**Presentation:** 3 good
**Contribution:** 2 fair
**Rating:** 5
**Confidence:** 4

**Summary:**

Under the federated learning paradigm, this work tries to validate the effectiveness of adding a momentum item to local client updating. Specifically, in each communication round, the proposed method will require the center server to broadcast the global gradient information to all clients (along with the updated model parameters). Then, the local clients embed the global gradient information into the local updating steps as a momentum item.

The strategy is simple and straightforward, just as the title highlighted. Particularly, this work gives detailed proof to validate the effectiveness of the simple strategy, which is appreciated (but with some mistakes). This work also empirically shows the effectiveness of the proposed method through some easy experiments.

**Strengths:**

The presentation is clear and logically consistent.
Furthermore, the proposed method, momentum acceleration by transiting the global gradient information to local clients, should be empirically useful and improve practical performance.

**Weaknesses:**

1.	The same algorithm has been proposed by multiple works [1,2]. FedCM in [1] is exactly the same as the proposed method, and [1] also gives theoretical guarantees under general convex or non-convex smooth assumptions, it is not the case as this work claimed: “However, whether momentum can offer theoretical benefits to FL remains underexplored”.

2.	There are mistakes with the proof of the main result (Theorem 1) and implicitly using of extra assumptions, so the correctness of the main results is doubtful. Please check the following Questions 2 and 4.

3.	The problem complexity is studied over the convergence rate under the same fixed hardware budget, such as communication cost or memory consumption. However, this work forgets to mention the extra requirement over the communication budget or the local storage.
(a)	Firstly, the costs of the downlink communication may be two times more expensive than other methods since the proposed method requires the broadcast of the extra averaged gradient information from the center server to each client, together with the new model parameters.
(b)	If the algorithm tries to maintain the same communication cost, then it can only broadcast the averaged gradient information. But it will require the local client to save a copy of the initial parameters of (r-1)-step, i.e., transferring the global updating step to local clients.

4.	Despite the doubt about proof, the method itself is not novel. Embedding local or global momentum seems straightforward. Meanwhile, considering the broad application scenarios of FL, the experiment is too simple, which is not valid to show the effectiveness of the method. However, if the proof can be validated, those shortages can be largely mitigated.

[1] Xu J, Wang S, Wang L, et al. Fedcm: Federated learning with client-level momentum[J]. arXiv preprint arXiv:2106.10874, 2021.
[2] Kim, Geeho, Jinkyu Kim, and Bohyung Han. "Communication-efficient federated learning with acceleration of global momentum." arXiv preprint arXiv:2201.03172 (2022).

**Questions:**

1.	I am totally not sure how to build the relationship between ∇fi(x) and ∇f(x) without the bounded data heterogeneity assumption. I.e., without the popular bounded data heterogeneity assumption, how the local gradient information contribute to the global convergence? Can you briefly explain it?

2.	The second question is independent but may be correlated with the first one. I checked the proof of Theorem 1 (Theorem 11 in the Appendix), and the building of the above relation can track the source back to Lemma 5. However, the very second inequality of the proof of Lemma 5 (Page-14), which builds the above relationship, seems to be wrong to me. Considering the updating step is x^{r+1} = x^{r} – γg^{r+1}, the item −γ||∇f(x^{r})||2 comes out from nowhere, and the sign of the item γ⟨∇f(x^{r}),g^{r+1}⟩ should be negative, and it is the real descent item.
Then, without Lemma 5, the most important result in this work, Theorem 1, may be wrong.

3.	This work uses Young’s inequality four times (Page 14, 17, 22 and 25), but actually, I am not really sure how to get the derived inequality by applying Young’s inequality each time. Please elaborate on each one.

4.	This question is correlated with the Question 1 and 3. Basically, each time when you use the Young’s inequality, you will use the relation, E||x^{r} – x{r−1}||^2 <= γ^{2}E(ξ_{r-1} + E||∇f(x^{r-1})||^{2}) in next step, for example, it has been used to derive the second inequality in Page 22.
It is not true to me without any assumptions. Basically, you are saying the averaged gradient can be bounded by the true gradient with some error, this is basically a (new) variant assumption of the assumption you claim you have abandoned (bounded data heterogeneity assumption). I believe you should not abandon the more standard bounded data heterogeneity assumption.

5.	I checked the mentioned work VRL-SGD that can handle unbounded data heterogeneity. It is not published, and the soundness of the proof is being doubted in the previously peer-reviewing stage. You should be careful to present the conclusions of VRL-SGD as formal results in your work.

---

> ### Author Response · Authors · 2023-11-20
> **Official Comment by Authors: Part I**
>
> We thank the reviewer for the comments. All questions have been clarified. We are glad to address any further comments or questions.
>
>
>
> **1. Comparison with FedCM**
>
>
> Thanks for pointing this out and bringing FedCM [R1] to our attention. It is indeed the same as our FedAvg-M. We will remove the statements like "resulting in the new algorithm FedAvg-M" from our paper. Please note that the main novelty of our paper does not lie in the algorithm development, but in **clarifying the theoretical improvements brought by momentum**. The FedCM paper cannot justify the theoretical benefits by incorporating momentum.
>
> To resolve the reviewer's concern, we added a detailed comparison with FedCM in the revised paper, see the highlighted paragraph in Page 6. For reviewer's convenience, we repeat the highlighted paragraph as follows:
>
> *FEDAVG-M coincides with the FEDCM algorithm proposed by Xu et al. (2021b). However, our results outperform that of Xu et al. (2021b) in several aspects. First, our convergence only utilizes the standard smoothness of objectives and gradient stochasticity while Xu et al. (2021b) additionally require bounded data heterogeneity and bounded gradients which are rarely valid in practice, suggesting the limitation of their analysis. Second, the convergence established by Xu et al. (2021b) is significantly weaker than ours and cannot even asymptotically approach the ideal rate $O(1/ \sqrt{NKR})$ in non-convex FL, as demonstrated by the results stated in Table 1.*
>
>
> The rate comparison established in FedCM and ours are as follows:
>
> FedCM: $\left(\frac{L\Delta ({\sigma^2}+NK{G}^2)}{NKR}\right)^{1/2}+\left(\frac{L\Delta ({\sigma}/{\sqrt{K}}+G)}{R}\right)^{2/3}$.
>
> FedAvg-M: $\left(\frac{L\Delta \sigma^2}{NKR}\right)^{1/2}+\frac{L\Delta}{R}$.
>
> It is observed that the rate established by FedCM is much weaker than FedAvg-M.
>
> **2. Mistakes in the proof**
>
> Please check the detailed responses to questions below.
>
> **3. The extra requirement over the communication budget or the local storage**
>
>
> First, it is generally true that downlink bandwidth is substantially larger than uplink bandwidth in most practical systems. This significant asymmetry between downlink and uplink capacities suggests that heavier downlink loads may be less problematic compared to lighter uplink loads.
>
> Second, our FedAvg-M does not incur additional downlink load compared to FedAvg when more memory is available. For practical implementation of FedAvg-M, the server does not need to communicate both $x^{r+1}$ and $g^{r+1}$ simultaneously to clients. If clients can store a model copy $x^r$ locally, by only broadcasting the latest server model $x^{r+1}$ from the server to all clients, clients are able to recover the momentum direction through $g^{r+1}=(x^{r+1}-x^r)/\gamma$. By employing the equivalent implementation, FedAvg-M does not suffer from a double in downlink communication.
>
> Moreover, we would like to remark that the doubling in local storage is customary for advanced FL algorithms beyond FedAvg, see, e.g., Scaffold [R2], Mime [R3]. In fact, as long as the local update rule is not the vanilla SGD, the double local storage can be inevitable to store optimizer states, e.g., momentum state.
>
> To resolve the reviewer's concern, we added comment in Section 3 of the revised paper. The comment is *"Notably, no extra downlink commmunication cost is needed if clients store the last iterate model $x^r$ so that momentum $g^{r+1}$ can recovered through $(x^{r+1}-x^r)/\gamma$."*
>
>
> **4. The method is not novel and the experiment is too simple**
>
> We respectively disagree with the reviewer on this point. The main novelty in our paper lies in **proving the theoretical improvements brought by momentum**, not in proposing new momentum variants of FL algorithms. While momentum has been widely used in federated learning empirically, it is still unknown in literature how momentum can benefit federated learning theoretically. Our paper focused on this open question and proved two novel results:
>
> - **Momentum can remove the influence of data heterogeneity**. When all clients participate in FL, momentum enables FedAvg to get rid of the restrictive data heterogeneity assumption without any impractical algorithmic structure.
>
> - **Momentum can theoretically speed up FL algorithm**. We have shown that FedAvg, Scaffold, and their variance-reduced variants with momentum achieve faster convergence rate than without momentum. In fact, all achieved results are state-of-the-art compared to existing algorithms.
>
> With the above two contributions, we are the **first**, to our best knowledge, to validate the usage of momentum in federated learning in theory, which we believe is a novel and important contribution to the community.

---

> ### Author Response · Authors · 2023-11-20
> **Official Comment by Authors: Part II**
>
> As for experiments, we have added a new experiment with $N=100$ clients with MNIST dataset in the appendix of the revision, see Figure 5 and 6. The results show that FedAvg-M and Scaffold-M outperform FedAvg and Scaffold by an evident margin when $N=100$, which is closer to the pratical FL setups.
>
> Also, since FedAvg-M coincides with FedCM. The superior performance of FedCM reported in FedCM paper is also an evidence that momentum can bring significant benefits, which aligns well with our theory.
>
> Finally, we would like to emphasize that the main contribution of our paper is to clarify the theoretical benefits brought by momentum. We hope the reviewer can find our empirical studies have well justified all our theoretical findings.
> ### Questions
> `Q1`: How to build the relationship between $\nabla f_i(x)$ and $\nabla f(x)$ without the bounded data heterogeneity assumption?
>
> A: We briefly present the high-level idea of FedAvg-M in the full participation setting. Note that when the local learning rate $\eta\to 0$ and global learning rate $\gamma$ is fixed, FedAvg will reduce to minibatch SGD over all the clients, i.e., SGD over $f(x)$ with batch size $NK$. In this case, the heterogeneity assumption $\zeta$ is known to be unnecessary. Intuitively, in FedAvg-M, our strategy is also to show that the local iterates will not drift too far away from the initialization in each communication round (Lemma 9) if $\eta$ is properly small with the help of momentum.
>
> `Q2`: Lemma 5 seems incorrect.
>
> A: As pointed out and confirmed by reviewer ELLH in the part of "Typos", the second line in Lemma 5 is merely a typo and **does not affect the overall validity**. The correction should be
> $$
> f(x^{r+1})\leq f(x^r)+\langle \nabla f(x^r),x^{r+1}-x^r\rangle +\frac{L}{2}\\|x^{r+1}-x^r\\|^2 = f(x^r)-\gamma  \\|\nabla f(x^r)\\|^2+\gamma  \langle \nabla f(x^r),\nabla f(x^r)-g^{r+1}\rangle+\frac{L\gamma^2}{2}\\|g^{r+1}\\|^2.
> $$
> We have addressed this in our revision.
>
> `Q3`: How to use Young inequality?
>
> A: In page 14, $\gamma  \langle \nabla f(x^r),\nabla f(x^r)-g^{r+1}\rangle \leq \frac{\gamma }{2}\\|\nabla f(x^r)\\|^2+\frac{\gamma }{2} \\|\nabla f(x^r)-g^{r+1}\\|^2$.
>
> In page 17 and 22, $\mathbb{E} [\\|\nabla f_i(x^r)\\|^2] \leq (1+q)\mathbb{E}[\\|\nabla f_i(x^{r-1})\\|^2]+(1+q^{-1})\\| \nabla f_i(x^r)-\nabla f_i(x^{r-1})\\|^2 \leq (1+q)\mathbb{E}[\\|\nabla f_i(x^{r-1})\\|^2]+(1+q^{-1})L^2\mathbb{E}[\\|x^r-x^{r-1}\\|^2]$.
>
> In page 25, $\left(1-\frac{S}{N}\right)\frac{1}{N}\sum_{i=1}^N\mathbb{E}[\\|c_i^r-\nabla f_i(x^r)\\|^2]\leq \left(1-\frac{S}{N}\right)\frac{1}{N}\sum_{i=1}^N\mathbb{E}\left[\left(1+\frac{S}{2N}\right)\\|c_i^r-\nabla f_i(x^{r-1})\\|^2+\left(1+\frac{2N}{S}\right)\\| \nabla f_i(x^r)-\nabla f_i(x^{r-1})\\|^2\right]$ and $\\| \nabla f_i(x^r)-\nabla f_i(x^{r-1})\\|^2 \leq L^2\\|x^r-x^{r-1}\\|^2$.
>
> `Q4`: About the second inequality in page 22 $\mathbb{E}||x^{r} – x^{r−1}||^2 \leq 2\gamma^{2}\mathbb{E}(\mathcal{E}_{r-1} + \mathbb{E}||\nabla f(x^{r-1})||^{2})$.
>
> A: First, we would like to emphasize that **we did not use any additional assumption**. Note that by the definition of $\mathcal{E}_{r-1}$ in the beginning of Appendix A, we have
>
> $\mathbb{E}\\|x^r-x^{r-1}\\|^2 = \gamma^2\mathbb{E}\\|g^r\\|^2 \leq 2\gamma^2 \mathbb{E}(\\|g^r-\nabla f(x^{r-1})\\|^2 + \\|\nabla f(x^{r-1})\\|^2) = 2\gamma^2 (\mathcal{E}_{r-1}+\mathbb{E}[\\|\nabla f(x^{r-1})\\|^2])$.
>
> And we derive a recursive bound for $\mathcal{E}_{r-1}$ in Lemma 8. Therefore, we do not need the bounded heterogeneity assumption.
>
> `Q5`: About VRL-SGD.
>
> A: Thanks for pointing this out and we have made additional comments on the work in our revision, as highlighted in the red color in Table 1. For reviewer's convenience, we repeat the comments as follows:
> *The works have not been published in peer-reviewed venues.*
>
> We hope the above clarification can relieve the reviewer's concern. We believe our paper makes a novel and important contribution to the community on understanding the theoretical benefits of momentum in federated learning, which has never been clarified before. We are happy to address any further questions or commmens from the reviewer.
>
> [R1] Xu, Jing, et al. ‘Fedcm: Federated Learning with Client-Level Momentum’. arXiv Preprint arXiv:2106. 10874, 2021.
>
> [R2] Karimireddy, Sai Praneeth, Satyen Kale, et al. ‘Scaffold: Stochastic Controlled Averaging for Federated Learning’. International Conference on Machine Learning, PMLR, 2020, pp. 5132–5143.
>
> [R3] Karimireddy, Sai Praneeth, Martin Jaggi, et al. ‘Mime: Mimicking Centralized Stochastic Algorithms in Federated Learning’. arXiv Preprint arXiv:2008. 03606, 2020.

---

> > ### Author Response · Authors · 2023-11-21
> >
> > Dear Reviewer W4SM,
> >
> > We hope our rebuttal has answered your questions and clarified the concerns. Please kindly let us know if there are additional questions we should address, before the interactive rebuttal system is closed.
> >
> > Thank you and happy thanksgiving.

---

> > > ### Comment · Reviewer_W4SM · 2023-11-21
> > > **Thanks for rebuttal**
> > >
> > > Dear authors,
> > >
> > > Thank you for the detailed rebuttal and it did resolve some of my concerns, and I would like to raise my score. However, I have some follow-up comments still.
> > >
> > > Given the definition $\xi_{r}: = \mathbb{E}[|| \nabla f(x^{r}) - g^{r+1} ||^{2}]$ in your proof and your replying, I believe it is a formal assumption instead of an “auxiliary variables to facilitate proofs”.
> > >
> > > For example, the very commonly used assumption bounded variance $\mathbb{E}[||g - \nabla f(x)||^{2}] \leq \sigma^{2}$, which is the key in the convergence analysis since it builds the relationship between stochastic gradient and ground-truth gradient. Your current definition potentially implies a similar assumption that the difference between stochastic gradient and ground-truth gradient is upper-bounded.
> > >
> > > A quick question: does the result still hold when $\xi = \infty$?
> > > I think your result does not hold when $\xi = \infty$, such as Lemma 8, so you are assuming a “bounded variance” assumption, which is basically similar to the standard one, bounded data heterogeneity.

---

> ### Author Response · Authors · 2023-11-22
>
> We thank the reviewer for the follow-up question. We use $\xi$ to denote the random data sampled per iteration within each client and use  $\mathcal{E}_r = \mathbb{E}[\|\nabla f(x^r) - g^{r+1}\|^2]$ throughout the manuscript. We believe the reviewer meant to ask for the clarification regarding the latter one.
>
>
> The reviewer has misunderstood the term $\mathcal{E}_r = \mathbb{E}[\\|\nabla f(x^r) - g^{r+1}\\|^2]$. **This term is NOT an assumption**. We do not need to assume the boundedness of $\mathbb{E}[\\|\nabla f(x^r) - g^{r+1}\\|^2]$. Instead, **its boundedness can be derived rigorously** buildng upon the standard objective smoothness (i.e., Assumption 1) and stochastic gradients (i.e., Assumption 3). Specifically, we have shown in the proof of Theorem 11 that
>
> $$
> \sum_{r=0}^{R-1}\mathcal{E}\_r\leq \frac{9}{7\beta }\mathcal{E}\_{-1} + \frac{2}{7}\mathbb{E} [\sum_{r=-1}^{R-2}\\|\nabla f(x^{r})\\|^2] + \frac{144}{7}(e\beta \eta KL)^2G_0 R+ \frac{36\beta \sigma^2}{7NK}R
> $$
>
> where $G_0=\sum_{i=1}^N\\|\nabla f_i(x^0)\\|^2/N$ and $\mathcal{E}_{-1} = \\|\nabla f(x^0) - g^{0}\\|^2$ are all constants by definition. By combining this with Lemma 5, we have
>
> $$\frac{1}{7} \sum_{r=0}^{R-1} \mathbb{E}\left[\left\\|\nabla f\left(x^r\right)\right\\|^2\right] \leq \frac{1}{\gamma} \mathbb{E}\left[f\left(x^0\right)-f^\star\right] +\frac{39}{56 \beta} \mathcal{E}_{-1}+\frac{78}{7}(e \beta \eta K L)^2 G_0 R+\frac{39 \beta \sigma^2}{14 N K} R,$$
>
> which reveals that $\mathbb{E} [\sum_{r=-1}^{R-2}\\|\nabla f(x^{r})\\|^2]/R$ is bounded provided with properly small  $\eta$ and $\beta$. This, in turn, reveals that $\sum_{r=0}^{R-1}\mathcal{E}_r/R$ is bounded is the averaged sense, which is sufficient to attain the ultimate convergence rate. **It is guaranteed that $\mathcal{E}_r$ cannot go to $+\infty$, in the averaged sense**.
>
>
> Again, We would also like to emphasize that no additional assumption regarding the *data heterogeneity*, i.e., $\sup_x \frac{1}{N}\sum_{i=1}^N\\|\nabla f_i(x)-\nabla f(x)\\|^2$, is needed in our paper due to the usage of momentum, which is a non-trival contribution to the community.
>
> We hope this can address your concern. We are glad to have further discussion on this point.

---

> > ### Author Response · Authors · 2023-11-22
> >
> > Dear Reviewer W4SM,
> >
> > We hope our rebuttal has answered your questions and clarified the concerns. As the interactive rebuttal system is scheduled to close within the next 17 hours, we would appreciate it if you can inform us of whether your concerns have been adequately addressed or if there are any additional questions we should attend to. Your prompt feedback is greatly appreciated.
> >
> > Thank you and happy thanksgiving.

---

### Official Review · Reviewer_CZuh · 2023-11-01

**Soundness:** 2 fair
**Presentation:** 2 fair
**Contribution:** 2 fair
**Rating:** 5
**Confidence:** 4

**Summary:**

This paper explores the application of momentum to enhance the performance of FEDAVG and SCAFFOLD, two leading federated learning algorithms. It achieves a faster convergence without relying on the bounded data heterogeneity assumptions and introduces new variance-reduced extensions, exhibiting state-of-the-art convergence rates.

**Strengths:**

1. This paper is easy to follow.
2. The incorporation of momentum enhances the convergence rates of both FedAvg and SCAFFOLD. And this improvement has been substantiated through both theoretical analysis and experimental validation.

**Weaknesses:**

1. The final convergence rate achieved by the authors does not sufficiently account for the impact of the momentum coefficient. Please clarify this issue.
2. In fact, FedDyn [1] demonstrates a faster convergence rate compared to the authors' findings in this paper, which is also without the need of clients’ variance assumptions. This observation may highlight the potential limitations in the author's theoretical contributions.
3. The authors' work seriously lacks comparative experiments, including comparison with various momentum-based federated algorithms [2].
4. The author's experimental work lacks a comprehensive discussion of a key hyperparameter, momentum coefficient.

[1] Acar, Durmus Alp Emre, et al. "Federated learning based on dynamic regularization."  ICLR, 2021.

[2] Reddi, Sashank, et al. "Adaptive federated optimization." ICLR, 2021.

**Questions:**

This paper lacks a comprehensive discussion regarding the limitations of the proposed algorithms. It is evident, for instance, that SCAFFOLD exhibits a suboptimal performance at very low sampling rates [2], leaving uncertainty regarding the extent to which the authors' improved algorithm can address this issue.

Post-rebuttal Comments:
I would like to thank the authors for their responses. Many of my concerns have been addressed, including both the theoretical and experimental analyses of the hyperparameter $\beta$. But I may still be a little concerned with the authors' claims on their theoretical contributions of the convergence, which is particularly defeated by the faster convergence rate achieved by FedDyn (though FedDyn requires a clients’ solution optimizer). Overall, I would raise my score to 5.

---

> ### Author Response · Authors · 2023-11-20
>
> We thank the reviewer for the comments. All questions have been clarified. We are glad to address any further comments or questions.
>
> **1. Influence of momentum coefficient**
>
> The momentum coefficient does not appear in the final convergence rate because we have **set up a concrete and proper value for $\beta$** when deriving the convergence rate. For example, when establishing the convergence of FedAvg-M in Theorem 11 in the appendix, we first prove that
>
> $$\frac{1}{R}\sum_{r=0}^{R-1}\mathbb{E}\|\nabla f(x^r)\|^2 = O\left(\frac{L\Delta}{R} + \frac{L\Delta}{\beta R} + \frac{\beta \sigma^2}{NK} + (\beta \eta K L)^2 G_0\right)$$
>
> which reflects how $\beta$ would influence the convergence rate. When we set $\beta = O(\sqrt{\frac{NKL\Delta}{\sigma^2 R}})$ and let $\eta \beta L$ be sufficiently small (see Theorem 11 for the value of $\eta \beta L$), we can simplify the convergence rate as
>
> $$\frac{1}{R}\sum_{r=0}^{R-1}\mathbb{E}\|\nabla f(x^r)\|^2 = O\left(\sqrt{\frac{L \Delta \sigma^2}{NKR}} + \frac{L\Delta}{R}\right)$$
>
> which is the SOTA convergence rate. Notably, we remark the above annealing $\beta$ is mainly set for theory purpose.
>
> In practice, we found a proper constant value (e.g., $\beta=0.5, 0.2$) is also capable to attain substantial improvement, as demonstrated by our experiments.
>
> **2. FedDyn and our algorithms are not comparable**
>
> We respectively disagree with the comments that FedDyn demonstrates faster convergence rate than our proposed algorithms. **FedDyn and our algorithm belong to different algorithm families, and they are using different oracles in local updates.** Our algorithm use the standard stochastic gradient oracle in local updates, while FedDyn relies on the black-box oracle that gives the exact solutions to local optimization problems (see line 5 in Algorithm 1 of FedDyn). Note that the local optimization problem is generally intractable (particularly for non-convex functions) and thus the oracle required by FedDyn is usually impractical, which outweighs its benefits in convergence if any. If the local optimization problem is not solved exactly, it is not known whether the current rate in FedDyn still hold. For this reason, we believe FedDyn and our algorithms are not comparable. The convergence rate of FedDyn does not hurt our theoretical contribution.
>
> **3. Empirical comparison with federated adaptive algorithms**
>
> We thank the reviewer for bringing up the federated adaptive algorithms. However, the main contribution in our paper is to clarify the theoretical improvements brought by introducing momentum to FL algorithms. This goal, to our knowledge, has not been achieved by [R1] or any other literature.
>
> Again, we are not aiming to develop adaptive algorithms that can outperform any federated adaptive algorithms in [R1] empirically. All of our experiments are conducted to validate our theoretical findings, i.e., momentum can help remove the influence of data heterogeneity and enhance the convergence rate. Whether our algorithms can outperform [R1] or not does not provide direct evidence to our theoretical findings. We hope the reviewer can understand that the comparison between our algorithms and adaptive algorithms in [R1] is not necessary and beyond the scope of our paper.
>
> **4. Experiments on different $\beta$**
>
> As illustrated in the descriptions of algorithms, when $\beta=1$, ours will reduce to vanilla FedAvg or Scaffold. And in the experiments, we have shown the difference between $\beta=1$ and other constants, i.e. between the non-momentum variant and the momentum variant. Also, we supplement an ablation study on different choices of $\beta\in (0, 1)$ in the appendix of revision. The results suggest that stronger momentum can lead to better practical performance, see Figures 7 and 8 in Section D.5 in the appendix.
>
> **5. Scaffold with low sampling rate**
>
> We thank the reviewer for the sharp observation. When the number of sampled clients $S$ is far less than the number of all clients $N$, i.e., $S \ll N$, all federated learning algorithms listed in Table 2 will suffer from slower convergence, see their rates in Table 2. It implies that slow convergence in the scenario with a low client-sampling ratio is a common issue in federated learning.
>
> However, incorporating momentum does bring some benefits. For example, incorporating momentum has improved Scaffold's convergence rate from $O(\sqrt{\frac{L \Delta \sigma^2}{S K R}} + \frac{L \Delta}{R}(N/S)^{2/3})$ to $O(\sqrt{\frac{L \Delta \sigma^2}{S K R}} + \frac{L \Delta}{R}(N^{2/3}/S))$, which quantitively illustrates that momentum indeed accelerates the rate.
>
> We hope the above clarification can relieve the reviewer's concerns. We believe our paper makes a novel and important contribution to the community on understanding the theoretical benefits of momentum in federated learning, which has never been clarified before.
>
> [R1] Reddi, Sashank, et al. ‘Adaptive Federated Optimization’. arXiv Preprint arXiv:2003. 00295, 2020.

---

> > ### Author Response · Authors · 2023-11-21
> >
> > Dear Reviewer CZuh,
> >
> > We hope our rebuttal has answered your questions and clarified the concerns. Please kindly let us know if there are additional questions we should address, before the interactive rebuttal system is closed.
> >
> > Thank you and happy thanksgiving.

---

> > ### Author Response · Authors · 2023-11-22
> >
> > Dear Reviewer CZuh,
> >
> > We hope our rebuttal has answered your questions and clarified the concerns. As the interactive rebuttal system is scheduled to close within the next 17 hours, we would appreciate it if you can inform us of whether your concerns have been adequately addressed or if there are any additional questions we should attend to. Your prompt feedback is greatly appreciated.
> >
> > Thank you and happy thanksgiving.

---

> > ### Author Response · Authors · 2023-11-23
> > **Can we have your response**
> >
> > Dear Reviewer CZuh,
> >
> > The discussion period will end very soon. Can we know whether our response has clarified your concerns?
> >
> > Thank you and happy thanksgiving.

---

### Official Review · Reviewer_ELLH · 2023-11-01

**Soundness:** 4 excellent
**Presentation:** 3 good
**Contribution:** 3 good
**Rating:** 8
**Confidence:** 4

**Summary:**

This paper studies the impact of adding a simple momentum term to standard Federated Learning algorithms (namely, FedAVG and SCAFFOLD) to mitigate client drift by "anchoring" local gradients closer to an estimate of the gradient of the global function computed at the server side. State-of-the-art convergence rates are obtained in the non-convex and smooth setting, without relying on the common assumption of bounded data heterogeneity. Variance-reduced extensions of the algorithms are also studied. Baseline empirical evaluations of the methods are provided with a three-layer MLP and a ResNet18 on CIFAR-10, hinting that the introduced momentum term does indeed help generalizing on test data.

**Strengths:**

* **SOTA CV rates:** State-of-the-art convergence rates are obtained for the introduced methods.
* **No data heterogeneity assumption**: The proof technique gets rid of the bounded data heterogeneity assumption, improving theoretical convergence rates and hinting that the method mitigates the impact of arbitrary data heterogeneity.
* **No additional uplink load**: The introduced momentum term is simple, its effect is intuitive to understand, and does not lead to any additional client to server communication.
* **Constant step-size:** If the training is sufficiently long ($R$ sufficiently high), the theoretical analysis allows for a constant step size for the stochastic gradients (contrary to vanishing ones standard in the literature).
* **VR variants:** Variance-reduced variants of the methods are presented and analyzed.

**Weaknesses:**

* **Algorithm not new**: Contrary to what is claimed in section 3.1 (*"resulting in the new algorithm FEDAVG-M"*), the added momentum is not new: FedCM [[1]](https://arxiv.org/pdf/2106.10874.pdf) is exactly the same algorithm as FedAVG-M, although their theoretical analysis does use the bounded heterogeneity assumptions. Comparison with FedCM rates is lacking in Table 1.
* **Surprising rates for the VR variants**: [[2]](https://link.springer.com/article/10.1007/s10107-022-01822-7) states that *"every randomized algorithm requires $\mathcal O \left( \frac{\Delta L \sigma}{\epsilon^3} + \frac{\Delta L}{\epsilon^2} + \frac{\sigma^2}{\epsilon^2} \right)$  oracle queries"*, however, the rate reported in Table 1 and Theorem 2 for FedAVG-M-VR seems to improve on this lower bound as a it leads to an oracle complexity of $\mathcal O \left( \frac{\Delta L \sigma}{NK \epsilon^3} + \frac{\Delta L}{\epsilon^2} \right)$. Setting aside the variance-reduction effect of running a distributed algorithm (leading to the $NK$ term), is it normal to get rid of the $\frac{\sigma^2}{\epsilon^2}$ term or am I wrongly worried ? (This question also holds for the VR version of SCAFFOLD-M)
* **Experiments seem light:** A value of $N=10$ is pretty low for the standards of the literature in Federated Learning (see, e.g., experiments in [1] where a value of $N=100$ is used), especially since a different behavior for the optimization algorithms can be expected at scale in the partial participation setting (see, e.g. [[3]](https://arxiv.org/abs/2102.02079 ) ). Are the runs averaged over several random seeds ?
* **Additional downlink load**: Although no additional client-to-server communication is necessary, the server-to-client communications are doubled in size with the addition of the momentum for FedAVG-M.
* **Lacking discussion on link between $R$ and $\beta$**: While a constant step-size can be considered for sufficiently high values of $R$, the direct corollary is that, before that regime arrives, Theorem 1 sets a value of $\beta=1$, meaning that the theory seems to predict that the momentum could only be used for sufficiently long training. However, experiments in Fig. 2 seem to show that using a momentum would help even if the training stopped early.



[1] Jing Xu and Sen Wang and Liwei Wang and Andrew Chi-Chih Yao, *FedCM: Federated Learning with Client-level Momentum*, ArXiv eprint 2106.10874, 2021.

[2] Arjevani, Yossi, Carmon, Yair, Duchi, John C., Foster, Dylan J., Srebro, Nathan and Woodworth, Blake. *Lower bounds for non-convex stochastic optimization*, Mathematical Programming, 2023.

[3] Li, Qinbin and Diao, Yiqun and Chen, Quan and He, Bingsheng. *Federated Learning on Non-IID Data Silos: An Experimental Study*, 2022 IEEE 38th International Conference on Data Engineering (ICDE).

**Questions:**

* Does the performances of adding a momentum scales to settings with greater values of $N$ ? (Or does scaling leads to a collapses as could be observed for SCAFFOLD, see Fig.10 of [[3]](https://arxiv.org/abs/2102.02079 ) )?
* Although the last sentence of section 1.2 states *"The analysis presented in this work distinguishes from [[4]](https://arxiv.org/abs/2305.15155)"*, [[4]](https://arxiv.org/abs/2305.15155) state in their paper that *"We also hope that our proof techniques can be useful to establish linear speedup for other classes of distributed methods, e.g, algorithms based on local training such as SCAFFOLD and ProxSkip without relying on data similarity assumptions."* Thus, it raises the question: how different is your analysis from [[4]](https://arxiv.org/abs/2305.15155) ?

**Typos:**

* after equation (1): *"represents a~n~ global gradient"*.
* Second line of the proof of Lemma 5: the scalar product seems to be missing a term, shouldn't it rather read $\gamma \langle \nabla f(x^r), \nabla f(x^r) - g^{r+1} \rangle$  ? (this does not impact the following lines)


**Final comment:**

I recognize the interest of the theoretical contributions of this paper, thus, I am ready to increase my score if my concerns concerning the convergence rates and the experiments are correctly addressed.

[4] Ilyas Fatkhullin and Alexander Tyurin and Peter Richtárik. *Momentum Provably Improves Error Feedback!* ArXiv eprint 2305.15155, 2023.

=== **After Rebuttal** ===

My concerns and questions were correctly addressed by the authors, I subsequently raise my score and recommend to accept this paper.

---

> ### Author Response · Authors · 2023-11-20
> **Official Comment by Authors: Part I**
>
> We thank the reviewer for the detailed comments. All questions have been clarified. We are glad to address any further comments or questions.
>
> **1. Comparison with FedCM**
>
> Thanks for bringing FedCM to our attention. It is indeed the same as our FedAvg-M. We will remove the statements like "resulting in the new algorithm FedAvg-M" from our paper. Please note that the main novelty of our paper does not lie in the algorithm development, but in **clarifying the theoretical improvements brought by momentum**. The FedCM paper cannot justify the theoretical benefits by incorporating momentum.
>
> To resolve the reviewer's concern, we added a detailed comparison with FedCM in the revised paper, see the highlighted paragraph in Page 6. For reviewer's convenience, we repeat the highlighted paragraph as follows:
>
> *FEDAVG-M coincides with the FEDCM algorithm proposed by Xu et al. (2021b). However, our results outperform that of Xu et al. (2021b) in several aspects. First, our convergence only utilizes the standard smoothness of objectives and gradient stochasticity while Xu et al. (2021b) additionally require bounded data heterogeneity and bounded gradients which are rarely valid in practice, suggesting the limitation of their analysis. Second, the convergence established by Xu et al. (2021b) is significantly weaker than ours and cannot even asymptotically approach the ideal rate $O(1/ \sqrt{NKR})$ in non-convex FL, as demonstrated by the results stated in Table 1.*
>
>
> The rate comparison established in FedCM and ours are as follows:
>
> FedCM: $\left(\frac{L\Delta ({\sigma^2}+NK{G}^2)}{NKR}\right)^{1/2}+\left(\frac{L\Delta ({\sigma}/{\sqrt{K}}+G)}{R}\right)^{2/3}$.
>
> FedAvg-M: $\left(\frac{L\Delta \sigma^2}{NKR}\right)^{1/2}+\frac{L\Delta}{R}$.
>
> It is observed that the rate established by FedCM is much weaker than FedAvg-M.
>
> **2. VR rates**
>
> Thanks for the sharp observation. **There is no mismatch between our rate and that in [R1]. The difference lies in the number of data batches used in the initialization stage** (i.e., the the number of data batches in the very first iteration). In the paper we set initial number of batches $B$ as $\Theta\left((NKR)^{1/3}(\frac{\sigma^2}{L\Delta})^{4/3}\right)$ for simplicity so we can get rid of the additional $\sigma^2/\epsilon^2$ term. On the other hand, if we set initial batch size as $\Theta(NKR)$ as constrained by the setup in [R1], our rate will match the lower bound stated in [R1].
>
> To resolve the reviewer's concern, we also considered the setting with initial batch size $\Theta(NKR)$ and derived the convergence rate, see Theorem 15 and Theorem 22 in the revised paper. We also list the new convergence rate in Table 1. For reviewer's convenience, we put the comparison in FedAvg-M-VR as follows:
>
>
> For initial batchsize $B=\Theta\left((NKR)^{1/3}(\frac{\sigma^2}{L\Delta})^{4/3}\right)$, the rate is $\left(\frac{L\Delta \sigma}{NKR}\right)^{2/3}+\frac{L\Delta}{R}$.
> For initial batchsize $B=\Theta(NKR)$, the rate is $\left(\frac{L\Delta \sigma}{NKR}\right)^{2/3}+ \frac{\sigma^2}{NKR}+\frac{L\Delta}{R}$.
>
>
> **3. More experiments**
>
> Thanks for the advice. Due to the hardware resource limitation and tight rebuttal deadline, we cannot afford an experiment with $N=100$ with Cifar-10 dataset at this stage. Instead, we have conducted experiments with $N=100$ clients with MNIST dataset in the appendix of the revision, see Figure 5 and 6. The results show that FedAvg-M and Scaffold-M outperform FedAvg and Scaffold by an evident margin when $N=100$.
>
> Also, since FedAvg-M coincides with FedCM. The superior performance of FedCM reported in FedCM paper is also an evidence that momentum can bring significant benefits, which aligns well with our theory. Finally, we would like to emphasize that the main contribution of our paper is to clarify the theoretical benefits brought by momentum. The empirical studies are mainly to confirm our theoretical findings.

---

> ### Author Response · Authors · 2023-11-20
> **Official Comment by Authors: Part II**
>
> **4. Downlink load**
>
> First, it is generally true that downlink bandwidth is substantially larger than uplink bandwidth in most practical systems. This significant asymmetry between downlink and uplink capacities suggests that heavier downlink loads may be less problematic compared to lighter uplink loads.
>
> Second, our FedAvg-M does not incur additional downlink load compared to FedAvg when more memory is available. For practical implementation of FedAvg-M, the server does not need to communicate both $x^{r+1}$ and $g^{r+1}$ simultaneously to clients. If clients can store a model copy $x^r$ locally, by only broadcasting the latest server model $x^{r+1}$ from the server to all clients, clients are able to recover the momentum direction through $g^{r+1}=(x^{r+1}-x^r)/\gamma$. By employing the equivalent implementation, FedAvg-M does not suffer from a double in downlink communication.
>
> To resolve the reviewer's concern, we added comment in Section 3 of the revised paper. The comment is *"Notably, no extra downlink commmunication cost is needed if clients store the last iterate model $x^r$ so that momentum $g^{r+1}$ can recovered through $(x^{r+1}-x^r)/\gamma$."*
>
> **5. Link between $R$ and $\beta$**
>
> In Theorem 1 we set $\beta=1$ when $R$ is small for simplicity. However, we remark that $\beta$, in principle, can be any $\Theta(1)$ constant that is less than $1$ instead of being exactly $1$. The convergence rates will not affected by this choice, which can be easily derived with very minor modifications to the proof.
>
> To resolve the reviewer's concern, we have revised the choise of $\beta$ in Theorems 1, 11, 15, 19 and 22.
>
> **6. Momentum scales with more clients**
>
> Thanks for raising this very interesting question. Based on our theories, the incorporation of momentum (once properly tuned) scales well when more clients participate in FL with the linear speedup term $O(1/\sqrt{NKR})$. With our new experiments with $N=100$, we do not observe the collapse of Scaffold-M. Once we have more computational recourse and allowed time, we will conduct more expereiments to examine it.
>
>
> **7. Difference from [R4]**
>
> [R4] focuses on communication compression in distributed optimization and does not involve any fundamental characteristic of FL, i.e. local updates and partial participation. Therefore, our analysis is fundamentally different from [R4]. On the other hand, we propose momentum to handle the effect of client drift in local iterations, which is beyond the scope of [R4]. We don't think there is a direct connection between our analysis and [R4] except for the usage of momentum in algorithmic development.
>
> **8. References**
>
> We thank the reviewer for bringing various relevant references to our attention. We have cited them properly and added corresponding discussions in the revised paper.
>
>
> [R1] Arjevani, Yossi, Carmon, Yair, Duchi, John C., Foster, Dylan J., Srebro, Nathan and Woodworth, Blake. Lower bounds for non-convex stochastic optimization, Mathematical Programming, 2023.
>
> [R2] Karimireddy, Sai Praneeth, Satyen Kale, et al. ‘Scaffold: Stochastic Controlled Averaging for Federated Learning’. International Conference on Machine Learning, PMLR, 2020, pp. 5132–5143.
>
> [R3] Karimireddy, Sai Praneeth, Martin Jaggi, et al. ‘Mime: Mimicking Centralized Stochastic Algorithms in Federated Learning’. arXiv Preprint arXiv:2008. 03606, 2020.
>
> [R4] Fatkhullin, Ilyas, et al. ‘Momentum Provably Improves Error Feedback!’ arXiv Preprint arXiv:2305. 15155, 2023.
>
> We thank the reviewer again for the careful and valuable comments. We hope these response can clarify the reviewer's questions. We are looking forward to the follow-up discussion, and more than happy to address any further comments or questions.

---

> ### Comment · Reviewer_ELLH · 2023-11-20
>
> I thank the authors for their detailed answers.
>
> Given that:
> * They recognized the existence of FedCM and added a comparison of convergence rates in Table 1, as well as added a discussion paragraph in their main paper,
> * They clarified my concern about their convergence rate of the VR versions, and added those clarifications in the paper,
> * They added more experiments to show that FedAVG-M is indeed helping at scale,
> * They corrected my statement by remarking that no additional downlink communications are needed for a small memory overhead on the clients' part,
> * They clarified my concern about the use of momentum for low training rounds $R$,
> * The fact that, upon a second quick check of the proof of their Theorem 1, I did not see any blatant error in their derivations, making me confident about the validity of their result,
>
> and, most importantly, given that the theoretical contributions of this paper on showing the ability of a simple momentum term to provably help federated optimization without relying on the bounded heterogeneity assumption seems to be of high interest for the community,
>
> I subsequently raise my score and support fully the acceptance of the paper.
>
> I would, however, point out that more experiments on CIFAR-10 with ResNets to discuss the scaling properties of the momentum versions of the algorithms as well as the proper tuning of it when scaling in less of a "toy scenario" would strengthen the paper.

---

> > ### Author Response · Authors · 2023-11-21
> >
> > We are delighted that your concerns have been resolved, and we sincerely appreciate your positive feedback. We will incorporate your suggestions to strengthen our experimental results in our later revision. Thank you again for your valuable input.

---

### Official Review · Reviewer_XBW2 · 2023-11-08

**Soundness:** 2 fair
**Presentation:** 3 good
**Contribution:** 2 fair
**Rating:** 5
**Confidence:** 4

**Summary:**

This paper focuses on the FedAvg and SCAFFOLD algorithm in federated learning. In literature, many works that analyze the performance of these two algorithms have to rely on bounded heterogeneity assumptions, which is unrealistic. In this works, momentum methods is employed to solve the data heterogeneity issue in federated learning. Without any other modification on Fedavg and SCAFFOLD, plain SGD with momentum updates can achieve similar convergence rate as literature. Furthermore, this work show that in the setting of partial client participation, momentum update can accelerate convergence.

**Strengths:**

1. This work overcomes one of the most common problem in FL analysis, the data heterogeneity issue. Although a lot of works in literature analyzes the convergence result of the two algorithm, most of the works have bounded heterogeneity assumptions. This is the most basic problem in FL analysis. This work utilizes momentum method to overcome the difficulty.
2. The experiment result is encouraging and directly validate the theory.

**Weaknesses:**

The major concern is novelty. FedAvg and SCALFFOLD are well-known methods in FL. Momentum method is also a popular optimization algorithm. Thus the algorithm design lacks novelty. Further, some work has analyzed the performance of FedAvg with Adam update, e.g, Reddi, Sashank, et al. "Adaptive federated optimization." arXiv preprint arXiv:2003.00295 (2020). Adam algorithm is closely related to SGD with momentum, thus the proposed analysis lacks novelty.

**Questions:**

What is the major difference or difficulty of SGD momentum analysis compared to Adam algorithm? Reddi, Sashank, et al. "Adaptive federated optimization." arXiv preprint arXiv:2003.00295 (2020)

---

> ### Author Response · Authors · 2023-11-20
>
> We thank the reviewer for the comments. All questions have been clarified. We are glad to address any further comments or questions.
>
> **1. Novelty**
>
> The main novelty in our paper lies in **proving the theoretical improvements brought by momentum**, not in proposing new momentum variants of FL algorithms. While momentum has been widely used in federated learning empirically, it is still unknown in literature how momentum can benefit federated learning theoretically. Our paper focused on this open question and proved two novel results:
>
> - **Momentum can remove the influence of data heterogeneity**. When all clients participate in FL, momentum enables FedAvg to get rid of the restrictive data heterogeneity assumption without any impractical algorithmic structure.
>
> - **Momentum can theoretically speed up FL algorithm**. We have shown that FedAvg, Scaffold, and their variance-reduced variants with momentum achieve faster convergence rate than without momentum. In fact, all achieved results are state-of-the-art compared to existing algorithms.
>
> With the above two contributions, we are the **first**, to our best knowledge, to validate the usage of momentum in federated learning in theory, which we believe is a novel and important contribution to the community.
>
>
> **2. Comparison with federated ADAM**
>
> Reference [R1] is a great and influential paper. However, **it does not provide sufficient theoretical justification for the benefits brought by momentum**. [R1] primarily focuses on proving convergence guarantees, but does not clearly explicate the advantages of using momentum and adaptability. Its analysis is limited in that it does not clarify whether and how momentum can relax restrictive assumptions or improve convergence rates. The major novelty in our analysis is to utilize new analytical techniques to clarify that momentum can relieve data heterogeneity assumption and speed up  convergence theoretically.
>
> There are also several important technical differences between [R1] and our paper.  First, the analysis in [R1] relies on the impractical and restrictive assumptions of bounded gradient and bounded client variance, which can significantly simplify their analysis. In our analysis, we have relaxed all these assumptions. Second, the algorithm in [R1] use vanilla SGD in local updates (see Algorithm 2 in [R1]), while our paper incorporates momentum in local updates, which also cause significant difference in analysis.
>
> We hope the above clarification can relieve the reviewer's concern on the novelty. We believe our paper makes a novel and important contribution to the community on understanding the theoretical benefits of momentum in federated learning, which has never been clarified before.
>
> [R1] Reddi, Sashank, et al. ‘Adaptive Federated Optimization’. arXiv Preprint arXiv:2003. 00295, 2020.

---

> > ### Author Response · Authors · 2023-11-21
> >
> > Dear Reviewer XBW2,
> >
> > We hope our rebuttal has answered your questions and clarified the concerns. Please kindly let us know if there are additional questions we should address, before the interactive rebuttal system is closed.
> >
> > Thank you and happy thanksgiving.

---

> > ### Author Response · Authors · 2023-11-22
> >
> > Dear Reviewer XBW2,
> >
> > We hope our rebuttal has answered your questions and clarified the concerns. As the interactive rebuttal system is scheduled to close within the next 17 hours, we would appreciate it if you can inform us of whether your concerns have been adequately addressed or if there are any additional questions we should attend to. Your prompt feedback is greatly appreciated.
> >
> > Thank you and happy thanksgiving.

---

> > ### Comment · Reviewer_XBW2 · 2023-11-22
> > **Reply to Authors**
> >
> > Thanks to the authors for the answers and clarification. I agree with the authors that momentum methods can speed up the convergence, but this is commonly known. However, I do not see any benefit specific to FL setting. (Momentum can benefit centralized training, and it benefits FL in similar way.) Further, momentum can 'smoothen' the optimization landscape, which is also well studied in literature. The idea behind this is that it can close the gap between the gradients, which is also the goal of algorithm design when data is heterogeneous in FL. Although it is proved in the paper that momentum methods enable the convergence without data heterogeneity, theoretically it still remains unclear how momentum brings the benefit. If the goal of the paper is to justify the benefit of momentum, I would suggest quantify and add more details to the "Intuition on the effectiveness of momentum" part.

---

> > > ### Author Response · Authors · 2023-11-23
> > > **Thanks for the follow-up comments**
> > >
> > > Dear Reviewer XBW2,
> > >
> > > Thanks very much for your follow-up comments. You might have some misunderstandings on the role of momentum in both centralized and FL setting. We are very glad to have more discussions on this point.
> > >
> > > First, we respectfully disagree that "momentum methods can speed up the convergence" is a commonly known fact. To our best knowledge, **momentum SGD in the non-convex centralized setting cannot outperform vanilla SGD in theoretical convergence rate** (though it can be faster empirically). Below are some evidences from literature:
> > >
> > > - [R1] K. Yuan, et.al., "On the influence of momentum acceleration on online learning", JMLR, 2016: it proves that momentum SGD is equivalent to vanilla SGD with enlarged learning rate when learning rate is sufficiently small.
> > >
> > > - [R2] Y. Arjevani, et. al., "Lower Bounds for Non-Convex Stochastic Optimization", Mathematical Programming, 2023: it establishes the lower bound of stochastic first-order algorithms, and vanilla SGD can achieve this lower bound. It implies that vanilla SGD has the optimal theoretical convergence rate, and momentum SGD cannot outperform it in theory.
> > >
> > > - [R3] S. Ganesh, et. al., "Does Momentum Help in Stochastic Optimization? A Sample Complexity Analysis", UAI, 2023: it establishes the lower bound for momentum SGD and proves that SGD can achieve such lower bound.
> > >
> > > - [R4] R. Kidambi et. al., "On the insufficiency of existing momentum schemes for Stochastic Optimization", ICLR, 2018: it proves that there exist simple problem instances where momentum SGD cannot outperform SGD despite the best setting of its parameters.
> > >
> > > - [R5] C. Liu and M. Belkin, "Accelerating SGD with momentum for over-parameterized learning", ICLR, 2020: it proves that momentum SGD with any parameter selection does not in general provide acceleration over ordinary SGD.
> > >
> > > We are not aware of results proving that momentum SGD, in the non-convex scenario, can have a faster theoretical convergence rate than vanilla SGD under standard assumptions and in the centralized setting. We will appreciate it if you can provide us some relevant references on this result.
> > >
> > > Second, the momentum benefit we established is **exclusively specific to FL**. As we discussed above, momentum cannot bring convergence benefits to vanilla SGD in the centralized setting. The reason why we can show momentum brings benefits to FL is that momentum can relieve the influence of data heterogeneity as well as clients partial participation, which is apparently specific to FL but not any other setting. This is a brand novel contribution, since we did not borrow any idea from momentum in centralized setting (because momentum does not bring theoretical benefits in this setting).
> > >
> > > Third, we have provided rigorous proof on the benefits of momentum which, we believe, has "theoretically" show how momentum brings the benefits. We will be very grateful if you can give us more concrete suggestions on showing how momentum can theoretically bring benefits.
> > >
> > > Thanks again for these discussions. We are happy to clarify any further questions or comments.

---

### Author Response · Authors · 2023-11-20
**Global Response: Part II**

**More experimental results**

We have conducted additional experiments to address the reviewers' concerns. Specifically, we have added a new experiment involving 100 clients to investigate the scalability of momentum incorporation, as shown in Figures 5 and 6. This experimental setup closely resembles practical federated learning scenarios. Additionally, we have examined how different momentum values impact empirical performance, as depicted in Figure 7, 8. Importantly, all of these experimental results are consistent with our theoretical findings.

Moreover, we would like to remind the reviewers that our contribution lies in establishing new theories on momentum. The purpose of our empirical studies is to validate our theoretical findings. We hope the reviewers can find the current experiments have well justified the benefits brought by momentum.

[R1] Reddi, Sashank, et al. ‘Adaptive Federated Optimization’. arXiv Preprint arXiv:2003. 00295, 2020.

---

### Author Response · Authors · 2023-11-20
**Global Response: Part I**

We thank all the reviewers for the valuable comments. All questions have been clarified. We are glad to address any further comments or questions. All revisions have been highlighted in the new submission. Here, we summarize the major comments and our responses for reviewer's convenience.


**The novelty in our theoretical contribution**


The main novelty in our paper lies in **proving the theoretical improvements brought by momentum**, not in proposing new momentum variants of FL algorithms. While momentum has been widely used in federated learning empirically, it is still unknown in literature how momentum can benefit federated learning theoretically. Our paper focused on this open question and proved two novel results:

- **Momentum can remove the influence of data heterogeneity**. When all clients participate in FL, momentum enables FedAvg to get rid of the restrictive data heterogeneity assumption without any impractical algorithmic structure.

- **Momentum can theoretically speed up FL algorithm**. We have shown that FedAvg, Scaffold, and their variance-reduced variants with momentum achieve faster convergence rate than without momentum. In fact, all achieved results are state-of-the-art compared to existing algorithms.

With the above two contributions, we are the **first**, to our best knowledge, to validate the usage of momentum in federated learning in theory, which we believe is a novel and important contribution to the community.


**Comparison with FedCM**

Thanks for pointing this out and bringing FedCM [R1] to our attention. It is indeed the same as our FedAvg-M. We will remove the statements like "resulting in the new algorithm FedAvg-M" from our paper. Please note that the main novelty of our paper does not lie in the algorithm development, but in **clarifying the theoretical improvements brought by momentum**. The FedCM paper cannot justify the theoretical benefits by incorporating momentum.

To resolve the reviewer's concern, we added a detailed comparison with FedCM in the revised paper, see the highlighted paragraph in Page 6. For reviewer's convenience, we repeat the highlighted paragraph as follows:

*FEDAVG-M coincides with the FEDCM algorithm proposed by Xu et al. (2021b). However, our results outperform that of Xu et al. (2021b) in several aspects. First, our convergence only utilizes the standard smoothness of objectives and gradient stochasticity while Xu et al. (2021b) additionally require bounded data heterogeneity and bounded gradients which are rarely valid in practice, suggesting the limitation of their analysis. Second, the convergence established by Xu et al. (2021b) is significantly weaker than ours and cannot even asymptotically approach the ideal rate $O(1/ \sqrt{NKR})$ in non-convex FL, as demonstrated by the results stated in Table 1.*


The rate comparison established in FedCM and ours are as follows:

FedCM: $\left(\frac{L\Delta ({\sigma^2}+NK{G}^2)}{NKR}\right)^{1/2}+\left(\frac{L\Delta ({\sigma}/{\sqrt{K}}+G)}{R}\right)^{2/3}$.

FedAvg-M: $\left(\frac{L\Delta \sigma^2}{NKR}\right)^{1/2}+\frac{L\Delta}{R}$.

It is observed that the rate established by FedCM is much weaker than FedAvg-M.

**Comparison with federated adaptive learning [R1]**

We kindly remind the reviewers that our paper is not aiming to develop adaptive federated learning algorithms, but to to clarify the theoretical benefits brought by incorporating momentum to existing well-known algorithms such as FedAvg and Scaffold. With this distinctive research purpose, the analysis and theoretical results in our paper is very different from [R1].


Reference [R1] **does not provide theoretical justification for the benefits brought by momentum**. [R1] primarily focuses on proving convergence guarantees of federated adaptive algorithms, but does not clearly explicate the advantages of using momentum and adaptability. Its analysis is limited in that it does not clarify whether and how momentum can relax restrictive assumptions or improve convergence rates. The major novelty in our analysis is to utilize new analytical techniques to clarify that momentum can relieve data heterogeneity assumption and speed up  convergence theoretically.

---

### Meta-Review · Area_Chair_nuTF · 2023-12-06

**Metareview:**

Summary:
This paper focuses on the FedAvg and SCAFFOLD algorithms in federated learning. In literature, many works that analyze the performance of these two algorithms have to rely on bounded heterogeneity assumptions, which is unrealistic. This work employs momentum methods to solve the data heterogeneity issue in federated learning. Without any other modification on Fedavg and SCAFFOLD, plain SGD with momentum updates can achieve a convergence rate similar to that of the literature. Furthermore, this work shows that momentum updates can accelerate convergence in partial client participation.

Strengths:
+ State-of-the-art convergence rates are obtained for the introduced methods.
+ No data heterogeneity assumption: The proof technique removes the bounded data heterogeneity assumption, improving theoretical convergence rates and hinting that the method mitigates the impact of arbitrary data heterogeneity.
+ No additional uplink load: The introduced momentum term is simple, its effect is intuitive to understand, and it does not lead to other client-to-server communication.
+ Constant step size: If the training is sufficiently long (sufficiently high), the theoretical analysis allows for a constant step size for the stochastic gradients (contrary to vanishing ones standard in the literature).
+ VR variants: Variance-reduced variants of the methods are presented and analyzed.
+ This paper is easy to follow.

Weaknesses:
- To the best of my understanding, the authors have adequately addressed most weaknesses.

**Justification For Why Not Higher Score:**

The paper's score lies on the border line of acceptance/rejection.
Also, being mostly theoretical, it is better to discuss this in front of a poster with the dedicated/interested group in the same topic.

**Justification For Why Not Lower Score:**

For the reasons mentioned above.

---

### Decision · Program_Chairs · 2024-01-16

Accept (poster)